# Evidence for causal top-down frontal contributions to predictive processes in speech perception

Thomas E. Cope[1], E. Sohoglu[2], W. Sedley[3], K. Patterson[1,2], P.S. Jones[1], J. Wiggins[1], C. Dawson[1], M. Grube[3], R.P. Carlyon[2], T.D. Griffiths[3], Matthew H. Davis [2] & James B. Rowe [1,2]

Perception relies on the integration of sensory information and prior expectations. Here we show that selective neurodegeneration of human frontal speech regions results in delayed reconciliation of predictions in temporal cortex. These temporal regions were not atrophic, displayed normal evoked magnetic and electrical power, and preserved neural sensitivity to manipulations of sensory detail. Frontal neurodegeneration does not prevent the perceptual effects of contextual information; instead, prior expectations are applied inflexibly. The precision of predictions correlates with beta power, in line with theoretical models of the neural instantiation of predictive coding. Fronto-temporal interactions are enhanced while participants reconcile prior predictions with degraded sensory signals. Excessively precise predictions can explain several challenging phenomena in frontal aphasias, including agrammatism and subjective difficulties with speech perception. This work demonstrates that higher-level frontal mechanisms for cognitive and behavioural flexibility make a causal functional contribution to the hierarchical generative models underlying speech perception.

[1] Department of Clinical Neurosciences, University of Cambridge, Cambridge CB2 0SZ, UK. [2] Medical Research Council Cognition and Brain Sciences Unit, University of Cambridge, Cambridge CB2 7EF, UK. [3] Institute of Neuroscience, Newcastle University, Newcastle NE1 7RU, UK. Matthew H. Davis and James B. Rowe contributed equally to this work. Correspondence and requests for materials should be addressed to T.E.C. (email: thomascope@gmail.com)

It has long been recognised that perception relies on the integration of sensory input with expectations based on prior knowledge or experience[1]. This can be instantiated in hierarchical generative models, which contain both top-down connections for priors or beliefs about sensory evidence, and bottom-up connections for prediction error. The layers of these hierarchical models represent progressively more abstract descriptions of the underlying sensory data[2, 3]. An influential implementation is known as predictive coding[4, 5], in which the top-down generative connections express predictions for expected sensory signals, while bottom-up processes pass forward prediction errors to update the model. This method of information transfer is highly efficient[6]. Neural models of predictive coding are well formalised, and we therefore conceptualise and interpret our study in this framework.

There is empirical evidence for predictive coding in health, for vision[7, 8], hearing[9–11] and the link between perception and action in motor control[12–14]. Furthermore, dysfunctional predictive coding mechanisms can explain a range of neurological and psychiatric phenomena, in schizophrenia[15], functional movement disorders[16, 17], alien limb syndrome[18], tinnitus[19] and hallucinations[20]. Although these disorders have been explained in terms of aberrant predictive coding, the functional consequences of degradation of the neural architecture responsible for generating top-down predictions are unknown. This is a critical and novel test for hierarchical models of perception, which motivates the following hypothesis: degeneration of top-down prediction mechanisms in frontal lobe should have a substantial impact on lower-level sensory responses in temporal lobe, and should impair perceptual function when prior knowledge and sensory input must be combined (Fig. 1).

We test this hypothesis in the context of speech perception. Speech is a natural domain in which to study prediction, as humans are able to exploit a wide variety of visual, contextual and semantic cues to improve perception, especially in difficult listening environments[21, 22]. Indeed, contradictory beliefs established by mismatching visual and auditory speech can lead to false perception[23, 24]. It is important to note that such multimodal integration can be modelled in terms of predictive coding regardless of whether or not visual information occurs before auditory information[25]; what is important is that auditory sensory predictions are set up based on information from prior experience, sentential context or sensory information from another domain. We exploited the importance of written text in supporting perception of degraded speech[26, 27]. There is evidence for left lateralised top-down information transfer[28] from frontal language[29] and motor speech[30, 31] regions to auditory cortex during speech perception; in predictive coding theory this top-down transfer generates prior expectations for speech content and explains how listeners combine prior knowledge and sensory signals during perception and perceptual learning[32, 33].

To assess the effects of disrupted predictions we studied patients with early non-fluent primary progressive aphasia (nfvPPA), which is associated with selective neurodegeneration of the frontal lobe language and motor speech areas[34], but preservation of temporal lobe auditory regions. Disordered speech output in nfvPPA is characterised by apraxia of speech and/or agrammatism[35]. In contrast to stroke aphasia the neural damage

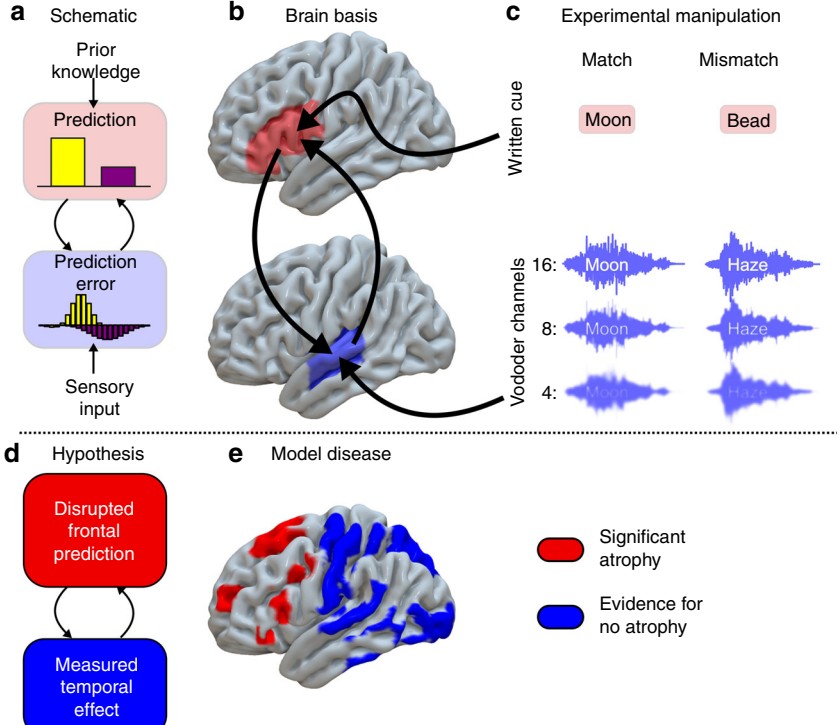

**Fig. 1** An illustration of the experimental motivation. **a** A schematic Bayesian framework for predictive coding in speech perception. **b** The putative brain basis of this framework[28]. Predictions are generated in inferior frontal gyrus and/or frontal motor speech regions (pink), and instantiated in auditory regions of superior temporal lobe (pale blue). **c** The two dimensional experimental manipulation employed here to detect a dissociation between normal temporal lobe responses to sensory detail (number of vocoder channels) and abnormal frontal lobe responses to prior congruency. **d** Our experiment relies on detecting the consequences of degraded predictions in abnormal frontal brain regions by measuring their effects in normal temporal regions. **e** Voxel-based morphometry in our patient group. Regions coloured in red displayed consistent reductions in grey matter volume (FWE $p < 0.05$). Regions coloured blue had strong evidence for normal cortical volume in nfvPPA (Bayesian probability of the null >0.7, cluster volume>1 cm$^3$). Uncoloured (grey) areas had no strong evidence for or against atrophy

**Table 1 Demographic details of the experimental groups**

|  | Number | Age | Gender | Age leaving education | MMSE | ACE-R | Raven's matrices |
|---|---|---|---|---|---|---|---|
| nfvPPA | 11 | 72 (9) | 8F, 3M | 18 (3) | 28 (2) | 84 (12) | 36 (9) |
| MEG controls | 11 | 72 (8) | 7F, 4M | 17 (2) | 29 (1) | 95 (2) | 46 (7) |
| MRI controls | 36 | 73 (7) | 17F, 19M | — | — | — | — |

Mean (standard deviation). There were no statistically significant differences in age, gender or education between nfvPPA patients and MEG controls. nfvPPA patients scored more poorly than controls on the Addenbrooke's Cognitive Examination (Revised) and Raven's Progressive Matrices, but were still within the population normal range. Most of the difference between nfvPPA and controls on the ACE-R was accounted for by verbal fluency. Audiometric thresholds are available in Supplementary Fig. 1B. One patient was unable to tolerate the MEG scanner environment so, for that case, results contribute only to the behavioural analysis

in frontal regions is partial[36, 37] enabling us to study a disruption of predictive mechanisms, rather than a system reorganised following their complete absence. Additionally, this patient cohort presents fewer problems for the modelling and interpretation of magnetoencephalography (MEG) or electroencephalography (EEG), as atrophy is subtle in early nfvPPA[38, 39].

For patients and matched control participants we recorded behavioural and neural data showing the influence of top-down and bottom-up manipulations on speech perception using an established paradigm involving presentation of written text that matches or mismatches with degraded spoken words[27, 32, 40]. With this paradigm, we can determine whether and how frontal cortical neurodegeneration impairs speech perception. The presence and function of top-down influences on speech perception is controversial (see refs. [41–44]), as is the question of whether frontal cortical regions make a critical contribution to speech perception, through predictive coding or alternative mechanisms. Some authors suggest that these contributions are task-specific and not a core component of speech perception systems[45]. The present study provides causal neural evidence with which to assess both of these claims.

Here we demonstrate distant neural effects of the degeneration of top-down signals from frontal lobes, during speech perception. We provide evidence of a direct relationship between the degree of frontal lobe degeneration and a delay in the neural mechanism for the reconciliation of predictions, which results in their inflexible application. Bayesian perceptual inference simulations demonstrate that this results in aberrantly precise prior expectations, which manifest as increased beta power during the instantiation of predictions, in agreement with theoretical frameworks of predictive coding. Finally, we show task-dependent enhancements in fronto-temporal interaction in nfvPPA, reflecting degraded neural mechanisms working harder to reconcile excessively precise predictions. We explain how inflexible predictions are able to account for several previously poorly understood symptoms and signs in frontal non-fluent aphasias, including difficulties with parsing the structure and content of running speech. Together, our results provide causal evidence for a critical role of frontal regions for the reconciliation of predictions during the perception and comprehension of speech.

## Results

**Structural consequences of nfvPPA.** To confirm the dissociation between frontal atrophy and intact temporal cortex, upon which our experiment relies, we used voxel-based morphometry to compare grey matter volume in nine of our patients with nfvPPA (see Table 1 for participant characteristics) to 36 age-matched healthy individuals using whole brain statistical parametric mapping (SPM) t-test and Bayesian null tests. As anticipated, brain regions of interest displayed a localised pattern of atrophy in nfvPPA (Fig. 1e), with grey matter volume loss in left inferior frontal regions (family wise error corrected (FWE) peak $p = 0.001$ at montreal neurological institute (MNI) [−37, 17, 7]; Bayes posterior probability of no difference <0.00001), but not in left

primary auditory cortex (FWE $p = 1$; Bayes posterior probability of no difference 0.75 at MNI [−59, −24, 9]) or superior temporal gyrus (FWE $p = 1$; Bayes posterior probability of no difference 0.91 at MNI [−67, −17, 3]). Significant atrophy was also observed in right inferior frontal regions (FWE $p = 0.004$; peak MNI [37, 20, 6]) but not right primary auditory cortex (FWE $p = 1$ at MNI [59, −24, 9]) or superior temporal gyrus (FWE $p = 1$ at MNI [67 −17 3]). Significant atrophy in left inferior frontal regions lay within pars triangularis, pars opercularis and anterior insula in the Desikan–Killiany Atlas (Supplementary Table 1).

**Subjective speech perception symptoms in nfvPPA.** While the core symptoms in nfvPPA relate to apraxia of speech and agrammatism, patients often complain of a feeling of speech deafness. To test for this symptom in our cohort, we asked patients and controls to rate their subjective difficulty with five listening scenarios, by placing a mark on a line from 'very easy' to 'very difficult'. Patients could respond appropriately to such rating scales. Patients and controls displayed very similar subjective difficulty ratings for 'speech in noise', 'localising sounds', 'understanding station announcements' and 'how loud others say their television is' (all $t(20)$ $p > 0.3$, Supplementary Fig. 1A). However, there was a difference in their assessments of difficulty in understanding speech in quiet environments ($t(20) = 2.66$, $p = 0.015$): controls universally rated this as very easy, while patients rated it to be almost as difficult as understanding speech in noise (interaction $F(1,20) = 8.21$, $p = 0.010$). Patients and controls had similar, age-appropriate, hearing acuity (Supplementary Fig. 1B).

**Evoked neural responses during the reconciliation of predictions.** To assess the neural correlates of degraded predictive mechanisms in nfvPPA (Fig. 1d), we recorded simultaneous MEG and EEG during a speech perception task. We manipulated prior expectations using matching or mismatching text cues, before participants heard spoken words that were varied in sensory detail by manipulating the number of vocoder channels (Figs. 1c, 2a)[46]. Overall evoked power was similar for the two groups of participants (Supplementary Fig. 3); frontal neurodegeneration did not lead to any large difference in the magnitude of the neural response evoked by single spoken words that could manifest as spurious group by condition interactions in neural activity.

To confirm that patients have normal responses to manipulations of sensory detail independent of predictions (as expected given preserved cortical volume in auditory cortex and superior temporal gyrus, Fig. 1e), we first assessed the neural effect of the number of vocoder channels (Fig. 3). Across all 21 individuals, SPM F-test peak effects were observed in the planar gradiometers at 96 ms (scalp-time FWE $p < 0.001$), in magnetometers at 188 ms (FWE $p < 0.001$) and again at 380 ms (FWE $p < 0.001$), and in EEG at 392 ms (FWE $p < 0.001$). These findings are consistent with previous studies in young individuals[40]. No reliable group by sensory interactions were found at the scalp locations of the peak main effect or at all scalp-time locations. Together, these results

are consistent with the idea that patients and controls produce similar neural responses to manipulations of sensory detail in degraded speech. Crucially for the interpretation of later results, the latency of these responses was also the same in both groups. In magnetometers, where the late effects of vocoder channel number were most clearly seen, control peaks occurred at 196 ms and 340 ms, and patient peaks at 172 ms and 372 ms (Fig. 3).

Having performed these control analyses, we tested our primary hypothesis by examining the neural effect of manipulating whether prior expectations matched sensory input. Across all 21 participants, during the reconciliation of predictions there

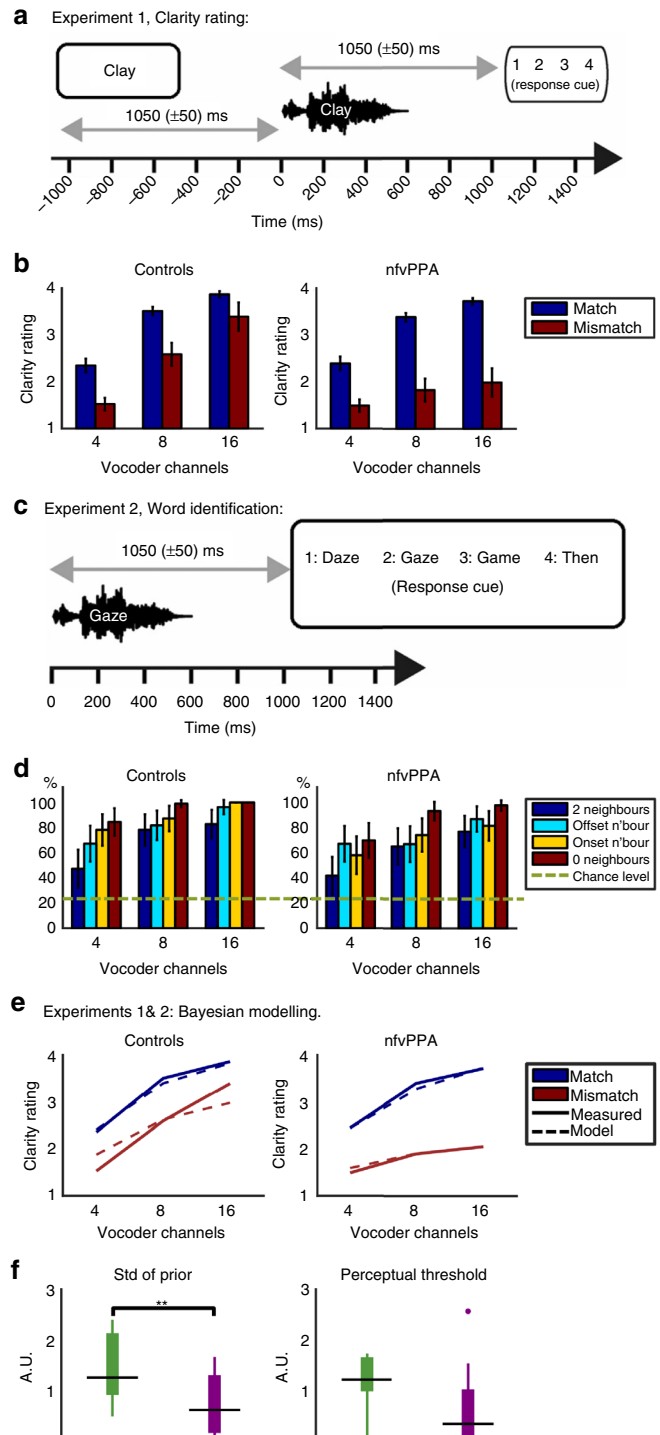

**a** Experiment 1, Clarity rating:

**b**   Controls              nfvPPA

Clarity rating
Vocoder channels
■ Match  ■ Mismatch

**c** Experiment 2, Word identification:

**d**   Controls              nfvPPA
■ 2 neighbours
■ Offset n'bour
■ Onset n'bour
■ 0 neighbours
-- Chance level

**e** Experiments 1& 2: Bayesian modelling.
Controls              nfvPPA
Clarity rating
Vocoder channels
■ Match
■ Mismatch
— Measured
-- Model

**f** Std of prior      Perceptual threshold
Controls  nfvPPA     Controls  nfvPPA

were significant SPM *F*-test main effects of cue congruency in all sensor types (Fig. 4a); planar gradiometers at 464 ms (scalp-time peak FWE $p < 0.001$), magnetometers at 400 ms (FWE $p < 0.001$), and EEG at 700 ms (FWE $p < 0.001$). At these scalp locations, significant group by congruency interactions were observed in the planar gradiometers and in the magnetometers (unpaired $t(19)$, $p < 0.05$ sustained over at least eight sequential samples), but not in the EEG electrodes. In the planar gradiometers, between 264 ms and 464 ms controls had a significantly larger effect of congruency than patients (Fig. 4b). The scalp topography averaged across this time window resembled that observed during the peak of the main effect, but with the pattern being stronger in controls (Fig. 4c). In the magnetometers, a cluster with similar timing and scalp topography was observed between 240 and 560 ms (Fig. 4b). Two additional clusters were also observed. In later time windows, from 728 to 808 ms, group by congruency interactions were observed in the opposite direction, with patients showing a significantly greater effect than controls. Again, the scalp topographies in this cluster resembled those during the main effect (Fig. 4c). This indicates that the effect of congruency was present in both groups, but that the effect was significantly delayed in nfvPPA. Finally, an early cluster was observed between 152 ms and 224 ms, with the controls displaying a significantly greater effect of congruency than patients. Intriguingly, the scalp topography in this time window was different to that observed during the conjoint main effect, with dipoles having a much more anterior centre of mass (Fig. 4c). This anterior topography is consistent with a frontal source, expected to appear in this earlier time window as shown in similar previous studies with young healthy listeners[33, 40].

To assess the underlying neural sources of these effects, multimodal sensor data were combined[47] and inverted into source space with sLORETA[48]. For the main effect of vocoder channel number, reconstructions were performed across all individuals combined, because no group difference or group by clarity interaction was demonstrated in sensor space. The main effect of sensory detail shown in MEG sensors and EEG electrodes is explained by increased activity for 16 channel speech in temporal lobe auditory areas in mid-latency time windows (200–280 ms and 290–440 ms; Fig. 5a), replicating previous findings in younger individuals[33, 40].

To localise the group by cue congruency interaction, we first display source reconstructions for the main effect of cue congruency in each group (Fig. 5b) for time windows defined by the main data features in overall sensor power averaged over conditions and participants (Supplementary Fig. 3). Given our findings of delayed congruency effects in patients, an additional, late, time window (710–850 ms) was also examined post hoc. We focus on the two principal sources observed in young healthy individuals for this task[33, 40], extracting average power in each

**Fig. 2** Behaviour. **a** Experiment 1 design. A Match trial is shown. In a Mismatch trial, the written and vocoded words would share no phonology (for example the written cue 'clay' might be paired with the vocoded word 'sing'). **b** Group-averaged clarity ratings for each condition. Error bars represent standard error across individuals within each group. **c** Four alternative forced choice vocoded word identification task. **d** Group-averaged per cent correct report for each condition. Chance performance at 25%. Error bars represent standard error across individuals within each group. **e** Overall group fits for single subject Bayesian data modelling of the data from **b**. **f** Derived parameters from the Bayesian data modelling. A.U., arbitrary units. Patients with nfvPPA displayed significantly more precise prior expectations than controls (Wilcoxon $U(11,11)$ $p < 0.01$). They also displayed a trend towards a reduction in perceptual thresholds (Wilcoxon $U(11,11)$ $p = 0.075$)

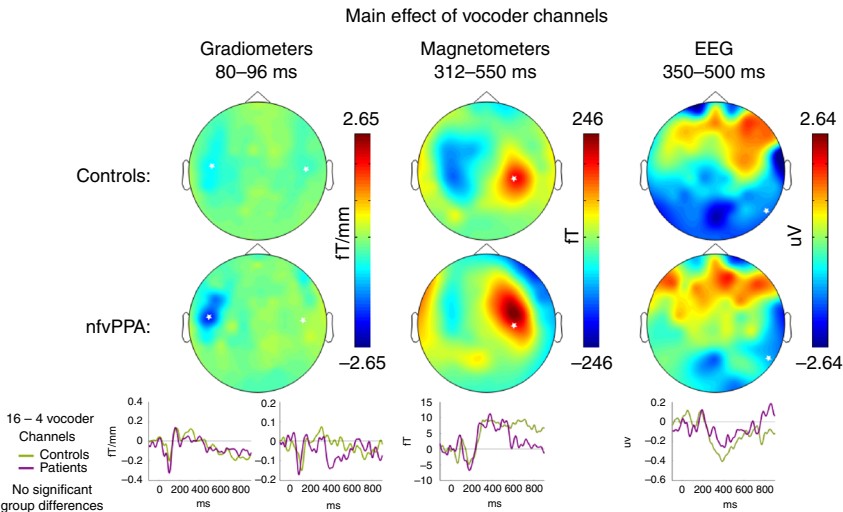

**Fig. 3** The effect of vocoder channel number. Illustrative topographic plots are shown of the main effect of vocoder channels across all participants. No group by sensory detail interactions were observed either at the peak locations (marked by white stars) or in a confirmatory SPM analysis

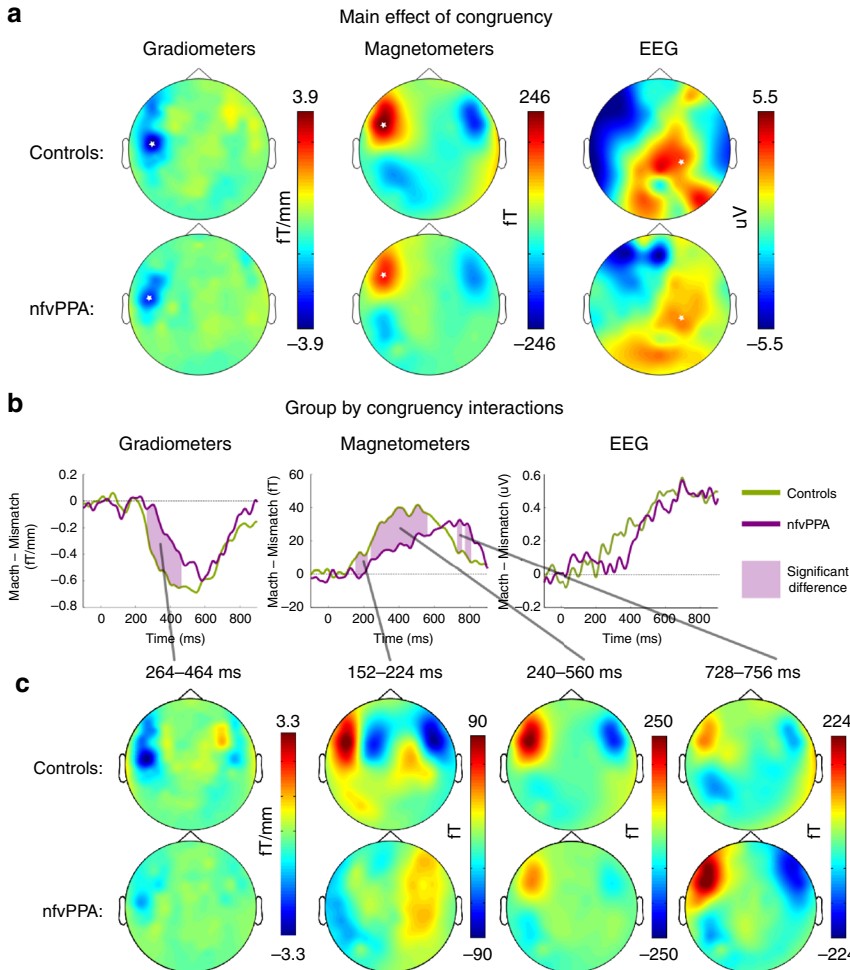

**Fig. 4** The effect of prime congruency. **a** Illustrative scalp topographic plots of the main effect of cue congruency for each group from 400 ms to 700 ms, a period of where both groups showed a large statistical effect of congruency with similar topography. White stars indicate the scalp location of the peak congruency effect across both groups between −100 ms and 900 ms (FWE $p < 0.001$ for all sensor types). **b** Significant group by congruency interactions ($p < 0.05$ sustained for more than 25 ms at the scalp locations marked by white stars in the upper panel) were observed in planar gradiometers and magnetometers, and are shaded in lilac. **c** Topographic plots for each group are shown averaged across each significant cluster of group by congruency interaction

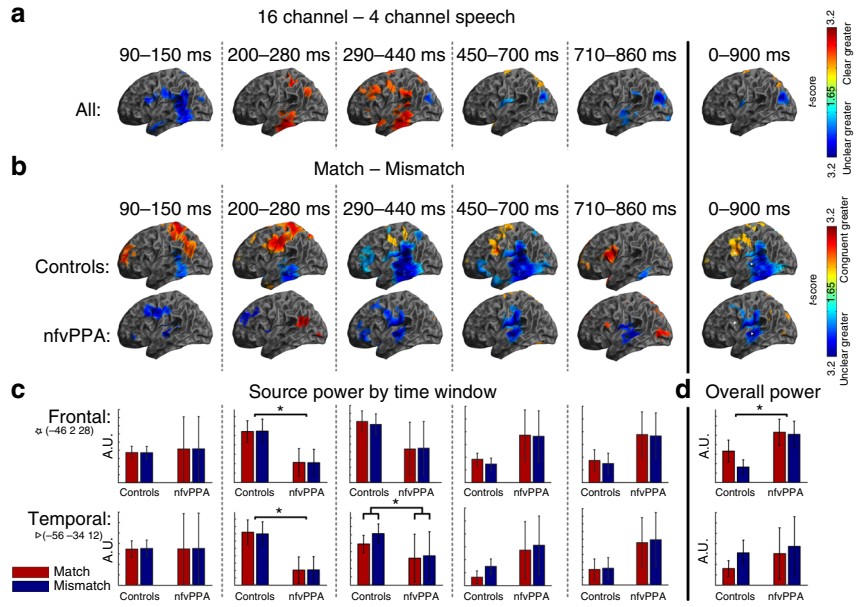

**Fig. 5** Evoked source space analysis. **a** Evoked source reconstructions (sLORETA) for the main effect of clarity for all participants combined. **b** Source reconstructions for the main effect of congruency for each group individually. **c** Illustrative bar charts are plotted at the bottom for source power by group and condition in the frontal (IFG) and temporal (STG) regions of interest for each time window. A.U., arbitrary units. Statistically significant differences are marked by asterisks (detail in 'Results' section of text). Error bars represent the between-subject standard error of the mean (not the between condition standard error, which is much lower due to the repeated measures design). **d** Overall power in each source by group and condition across the whole time window of interest

| A: Experiment 1 | DF | F | p (Greenhouse-Geisser) |
|---|---|---|---|
| **Group** | **1,40** | **7.7** | **0.011** |
| **Clarity** | **2,40** | **69.3** | **<0.001** |
| **Group * Clarity** | **2,40** | **17.1** | **<0.001** |
| Congruency | 1,40 | 1.2 | 0.287 |
| **Group * Congruency** | **1,40** | **13.2** | **0.002** |
| Clarity * Congruency | 2,40 | 2.3 | 0.140 |
| **Group * Clarity * Congruency** | **2,40** | **8.2** | **0.007** |
| **B: Experiment 2** | | | |
| **Group** | **1,120** | **10.9** | **0.003** |
| **Vocoder channels** | **2,120** | **5.8** | **0.015** |
| Distractor difficulty | 3,120 | 0.8 | 0.457 |
| Group * Clarity | 2,120 | 0.2 | 0.765 |
| Group * Distractor difficulty | 3,120 | 1.1 | 0.334 |
| **Vocoder channels * Distractor difficulty** | **6,120** | **4.2** | **0.004** |
| Group * Vocoder channels * Distractor difficulty | 6,120 | 1.8 | 0.149 |

**Table 2 Repeated measures ANOVA of behavioural data from experiments 1 (Fig. 1b) and 2 (Fig. 1d)**

Statistically significant rows at p<0.05 are indicated in bold

condition at frontal and superior temporal voxels of interest defined by the main effect of congruency averaged across the whole epoch (Fig. 5d). No group differences were demonstrated in the earliest (90–150 ms frontal $F(1,18) = 0.06$, $p = 0.81$, temporal $F(1,18) < 0.01$, $p = 0.99$) or later (450–700 ms frontal $F(1,18) = 4.24$, $p = 0.054$, temporal $F(1,18) = 2.84$, $p = 0.11$, 710–860 ms frontal $F(1,18) = 3.10$, $p = 0.10$, temporal $F(1,18) = 2.96$, $p = 0.10$) time windows. Between 200 and 280 ms, there were significant main effects of group in both frontal ($F(1,18) = 6.37$,

$p = 0.02$) and temporal ($F(1,18) = 5.07$, $p = 0.04$) voxels, with greater responses in controls than patients. Between 290 and 440 ms, this main effect had dissipated (frontal $F(1,18) = 2.51$, $p = 0.13$, temporal $F(1,18) = 1.63$, $p = 0.21$), but there was a group by condition interaction, with controls showing a greater effect of cue congruency in the superior temporal ($F(1,18) = 4.46$, $p = 0.049$) but not the frontal ($F(1,18) = 0.76$, $p = 0.39$) voxel.

The analysis across the whole epoch is of particular interest. Across all individuals, a repeated-measures ANOVA (Supplementary Table 2) confirmed the pattern of opposing effects of prior knowledge in frontal and superior temporal regions seen in a previous study[40] ($F(1,134) = 60.1$, $p < 0.001$). However, there was a group by source by congruency interaction ($F(1,134) = 11.9$, $p = 0.001$), primarily driven by the absence of a significant effect of congruency in frontal regions in nfvPPA (Fig. 5b). When the total power in the frontal region was examined, a main effect of group was observed such that patients had significantly more frontal power than controls, but their modulation of frontal power by congruency was absent (Fig. 5d).

**Behavioural experiment 1 vocoded word clarity rating.** Given these neural differences, we sought to understand the perceptual correlates of neural delay in the reconciliation of predictions by examining the behavioural consequences of manipulations of prior knowledge in our two groups (Figs. 2a, 1c). All individuals reported that the perceptual clarity of vocoded words was significantly increased by matching text cues (Fig. 2b), but this effect was greater in patients with nfvPPA than in controls. A repeated-measures ANOVA revealed that Group, Number of Vocoder Channels and Cue Congruency were significant either as main effects or as part of two-way or three-way interactions (see Table 2, Experiment 1 for statistical details).

A replication experiment outside the MEG scanner confirmed that the difference between match and mismatch trials was due to a facilitatory effect of matching prior knowledge and not simply increased confusion in the face of mismatching priors: ratings of

perceptual clarity after a mismatching text cue were not statistically different from those after a 'neutral' or uninformative cue (repeated measures ANOVA $F(1,120) = 2.09$, $p = 0.15$). Furthermore, patients with nfvPPA had a much larger difference in clarity rating between 'neutral' and 'match' trials than controls (Supplementary Fig. 1C). It is important to note that participants were explicitly instructed to rate clarity across their own range of perceptual experience within the experiment, and were given training until they were able to do this. Comparing clarity ratings across groups is not, therefore, a direct measure of comparative listening difficulty as a rating of '1' simply means 'one of the least clear words I heard in the experiment', while a '4' means 'one of the clearest words I heard'. To fully assess the perceptual basis of our findings, we assessed the elements contributing to perceptual clarity with a further experiment and Bayesian modelling. These elements included: (1) patients' and controls' ability to identify degraded spoken words and (2) participants' introspective ability to perform higher-level estimation of the global precision of sensory input.

**Behavioural experiment 2 vocoded word identification**. To ensure that our finding was not a consequence of impaired word identification in patients leading to a group difference in reliance on prior knowledge[49], we performed a second experiment in which participants identified noise vocoded words in the absence of prior expectations (Fig. 2c). To reduce response demands for patients with non-fluent speech we used a four-alternative forced-choice identification task. All individuals with nfvPPA were above chance at identifying even the most degraded vocoded speech and, as a group, performed almost as well as controls (Fig. 2d). Both groups were influenced in the same way by the number of noise vocoder channels and the number of close distractor items presented as alternatives in the forced choice. As expected, it was easier for all individuals to identify words with more vocoder channels and if there were fewer close distractor items. This effect was strongest for the most degraded speech, manifesting as an interaction between vocoder channels and distractor difficulty (Table 2, Experiment 2).

Crucially, these data show that a lower-level impairment in perceiving vocoded speech cannot be the sole explanation of our finding of an increased congruency effect in nfvPPA patients. Patients performed better at identifying speech with eight channels than controls did with four channels (repeated measures ANOVA $F(1,63) = 7.1$, $p = 0.015$). Yet, patients still display a larger congruency effect for 8-channel vocoded words than controls do for 4-channel speech ($t(20) = 2.17$, $p = 0.04$). Hence, the magnitude of congruency effects in clarity rating is not simply related to objective abilities at word identification, but rather reflects a difference in the mechanisms by which prior knowledge influences lower-level perceptual processing. We investigate the nature of this effect with a Bayesian perceptual model combining word report and clarity rating data.

**Bayesian modelling of experiments 1 and 2**. To dissociate changes in the precision of predictions from difficulties with higher level estimation of the precision of sensory input, we performed hierarchical Bayesian inference simulations (c.f. ref. [33]; Supplementary Fig. 2). Individual differences in word discriminability were accounted for by defining the precision of sensory input for each subject as the percentage above chance for word identification at each vocoder channel number in Experiment 2. This allowed us to individually optimise two free parameters against the clarity ratings measured in Experiment 1. These parameters were the precision of prior expectations (as measured by their standard deviation), and a perceptual threshold below which the observer rated speech as unclear (Supplementary

Fig. 2). The model explained 97.6% of the variance in the group-averaged clarity ratings (Fig. 2e). 99.4% of the variance could be explained by additionally accounting for non-linearities in the effect of the increasing sensory detail on perceptual clarity beyond 16 vocoder channels, but analysis of the Akaike information criterion suggested that this increase in variance explained did not outweigh the loss of parsimony compared to the simpler model (see Supplementary Discussion). The simpler model was therefore retained, but all of the group differences and associations between model outputs and neurophysiology reported in the results that follow remained significant if the complex model were used.

Patients had significantly more precise prior expectations than controls (Wilcoxon rank sum $U(11,11) = 83$, $p = 0.005$; Fig. 2f). There was a trend towards patients having lower perceptual thresholds ($U(11,11) = 99$, $p = 0.075$), meaning that patients required less sensory detail to give a clarity rating of 2 or higher, reflecting an appropriate downwards extension of the subjective clarity scale rather than a higher level introspective deficit resulting in patients not being ideal observers of their sensory experience (see 'Discussion'). The model results confirm that the consequence of degraded neural mechanisms for sensory predictions is not that the brain is unable to use prior knowledge (written cues) to modulate perception, but rather that patients with nfvPPA apply their prior knowledge with greater precision and inflexibility.

**Induced oscillatory dynamics**. To examine the effects of nfvPPA and task manipulations on induced oscillatory activity, we performed a time–frequency analysis of the planar gradiometer data, averaged across sensors (Fig. 6). First, we inspected the neural instantiation of predictions by analysing induced activity during the period following presentation of the written word but before the onset of the auditory stimulus. Based on recent studies in the auditory domain we expected this updating of predictions to manifest as an increase in beta frequency oscillations preceding the onset of the spoken word[19]. This was confirmed by SPM analysis across time–frequency space in our cohort, with a significant increase in beta power (10–28 Hz) for both groups of participants beginning around 800 ms after the onset of the written word, i.e. a~250 ms before the onset of the spoken word (cluster FWE $p = 0.001$, Fig. 6). At the time (992 ms) and frequency (24 Hz) of the peak effect for both groups overall, the single subject magnitude of the induced response correlated significantly with their precision of prior expectations as simulated by our behavioural Bayesian model (Pearson's $r(19) = -0.52$, $p = 0.017$; Spearman's $\rho = -0.54$, $p = 0.012$). A confirmatory SPM across the whole time window confirmed the group difference implied by this relationship, with patients displaying a single cluster of greater induced beta (20–34 Hz) power from 868 ms (cluster FWE $p = 0.010$). There were no induced effects that were greater in controls than in patients.

Second, we complemented our evoked analysis by assessing oscillatory power during the reconciliation of predictions, i.e. after the onset of the spoken word, averaged across sensors. The results in this section were not altered by subtraction of the condition-averaged evoked waveform subtracted from every trial, confirming that these are true induced responses rather than high-frequency contamination from the evoked responses described previously. Figure 7a illustrates the oscillatory power induced by hearing noise vocoded speech for each group, normalised to the pre-visual stimulus baseline and averaged across the whole brain. Across all conditions, the general pattern was for increased alpha and beta power for the first ~200 ms, followed by a desynchronisation from ~200 ms onwards.

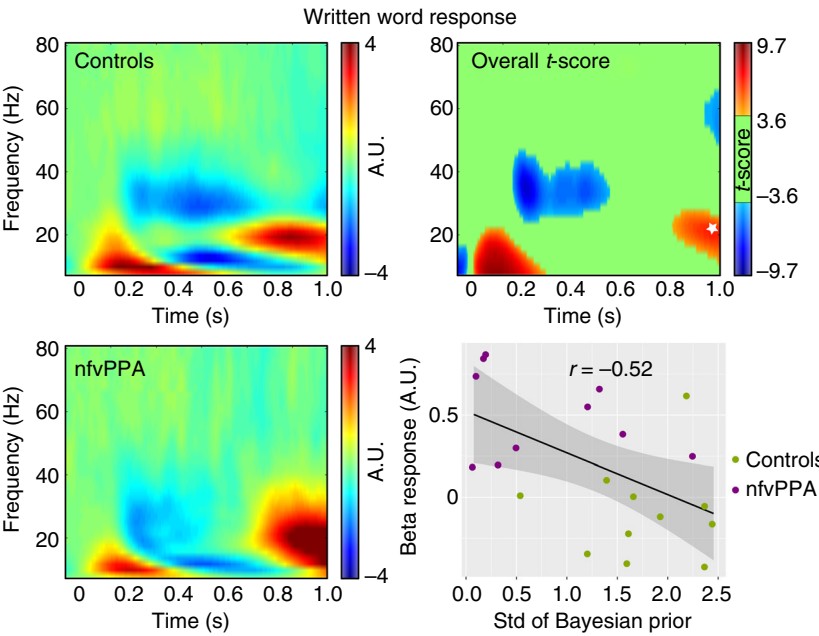

**Fig. 6** Analysis of induced responses after the written but before spoken word. Total induced power after written word onset by group, the overall *t*-score for difference from baseline for both groups combined, and the relationship between single subject response at the late beta peak (indicated by a white star) and precision of their prior expectations in the Bayesian behavioural model. A.U., arbitrary units. The grey shaded area in the bottom right plot indicates the 95% confidence band for the regression line, marked in black

SPM across time–frequency space demonstrated a significant main effect of congruency in both groups separately (Fig. 7a). In controls, this effect peaked at 436 ms at a frequency of 12 Hz (FWE $p < 0.001$), with greater suppression of this response for spoken words that matched prior written text. In patients, a similar effect was observed, also at 12 Hz, but with a later peak at 824 ms (FWE $p < 0.001$).

Across all 21 individuals SPM demonstrated two time–frequency clusters that showed a group by congruency interaction. In the first, extending from 276 to 444 ms between 4 and 24 Hz (peak 300 ms, 16 Hz), controls displayed a greater effect of congruency than patients (cluster FWE $p < 0.001$). In the second, beginning at 680 ms and extending beyond the end of the analysis from 6 to 34 Hz (peak 888 ms, 20 Hz), the interaction was reversed, with patients showing a greater effect of congruency than controls (cluster FWE $p < 0.001$). The scalp distribution and source localisations of this effect are illustrated in Supplementary Fig. 4: effects were restricted to the left hemisphere and localised to areas around superior temporal gyrus. Both interaction clusters remained significant (FWE $p \leq 0.002$) in a confirmatory analysis in which power was normalised to the pre-auditory stimulus baseline. To illustrate the time-course of this oscillatory dissociation, the data from all three sensor types were restricted to 12–24 Hz, encompassing the interactions in both directions, and the total effect of congruency on power in this band across the whole brain was plotted in Fig. 7b.

To investigate this delay in this beta response in individual patients, single subject time–frequency decompositions were performed and the time taken to reach 80% of the peak overall power contrast between matching and mismatching prior knowledge was defined for each subject. The effect latencies for controls were all tightly clustered between 275 and 400 ms (Fig. 7c). Every single patient was delayed compared to every single control, with a range of 412–1048 ms (Fig. 7d). In the patient group neural response latency was negatively correlated with grey matter volume in our left frontal region of interest ($r = -0.68$, $p = 0.042$; Fig. 7e) but not in our left temporal region of

interest ($r = 0.34$, $p = 0.36$; Fig. 7f). There was a trend towards a negative relationship between latency and the standard deviation of prior expectations, though this did not reach significance ($r = -0.37$, $p = 0.1$).

To summarise, all participants show the same congruency-induced reduction in activity in the STG, but nfvPPA patients are delayed in showing this response compared to controls. Thus, the differential response of patients and controls reflects a top-down effect of frontal neurodegeneration on brain responses in posterior regions that remain structurally intact (compare, Figs. 5b and 1e) and that respond normally to bottom-up manipulations of speech clarity (Fig. 3).

**Coherence and connectivity during prediction reconciliation.** To determine whether these effects are due to frontal degeneration or fronto-temporal disconnection, we examined coherence and connectivity between the frontal and temporal lobe sources of interest (Fig. 5c) during the 900 ms immediately following the onset of each spoken word. We employed two MEG connectivity analysis methods that give complementary information concerning fronto-temporal dynamics during the reconciliation of predictions: Imaginary Coherence, which is immune to volume conduction effects and source spread[50] as well as differences in power[51], and Grainger causality. These analyses allow us to be confident that relationships we describe are true reflections of the underlying brain dynamics. Both groups had significant fronto-temporal coherence up to around 25 Hz (Fig. 8a). Coherence in the beta band (13–23 Hz) was significantly stronger in patients than controls (Fig. 8b). Therefore an overall reduction of fronto-temporal connectivity cannot explain our observed differences between nfvPPA patients and controls.

Imaginary coherence does not provide robust indices of directionality, because inter-regional interactions potentially occur over more than one oscillatory cycle. We therefore also examined Granger causal relationships between our sources of interest[52]. This metric allows us to look at the directionality of

non-zero-lag fronto-temporal interactions while still being relatively robust to volume conduction. Grainger causality tests whether information from the past activity of one region can predict future activity in another better than its own past. Highly significant bi-directional Granger causal relationships were observed between temporal and frontal sources (Fig. 8c). To

compare these, while avoiding confounds due to differences in signal to noise ratio between regions and between individuals (which can alter the magnitude of Granger Causal relationships[50]), we divided the magnitude of each frequency value by the across-frequency mean for each individual and region to create a profile of relative influence for each region at each frequency. This analysis demonstrated significantly stronger temporal to frontal Granger Causal relationships at low frequencies, while frontal to temporal influences were stronger at higher, beta band, frequencies (Fig. 8d). These findings are in agreement with a recent study of MEG connectivity during written language comprehension, which showed that rhythmic information from temporal and parietal lobes was carried at lower frequencies than that from frontal cortex[53].

Overall, our finding of increased imaginary coherence in the patient group in a frequency band where frontal to temporal Granger causal influences predominate demonstrates that frontal neurodegeneration increases rather than reduces top-down connectivity from frontal to temporal regions during speech perception.

## Discussion

The principal finding of this study is that neurodegeneration of the frontal language network results in the delayed neural resolution of predictions in temporal lobe. Conversely, the temporal lobe neural responses to bottom-up manipulations of sensory detail were not delayed. This proves that the resolution of sensory predictions is causally mediated by frontal regions in humans. Our source-space analysis demonstrates that frontal regions are working harder overall in nfvPPA patients. Our finding that, in the patient group, coherence is increased in a frequency band where frontal to temporal influences predominate, suggests that increased fronto-temporal interaction is required to reconcile excessively precise predictions. This view is supported by recent observations that left inferior frontal imaginary coherence is decreased in nfvPPA during the resting state[54], confirming that the increased fronto-temporal coherence we observe here reflects the specific engagement of these mechanisms during language perception and not simply a global upregulation. Together, the results of this study resolve a key controversy in speech perception by demonstrating that frontal regions play a core role in reconciling predictions with sensory input during speech perception[28–30].

This observed impairment of predictive processing has significant perceptual consequences. Most strikingly, frontal neurodegeneration does not reduce the degree to which the brain employs contextual prior knowledge to guide lower-level speech perception. Rather, through Bayesian modelling, we show that prior knowledge of expected speech content is applied in an overly precise or inflexible fashion, thereby producing a larger-than-normal behavioural effect of prior knowledge on nfvPPA patients' ratings of speech clarity. In validation of computational

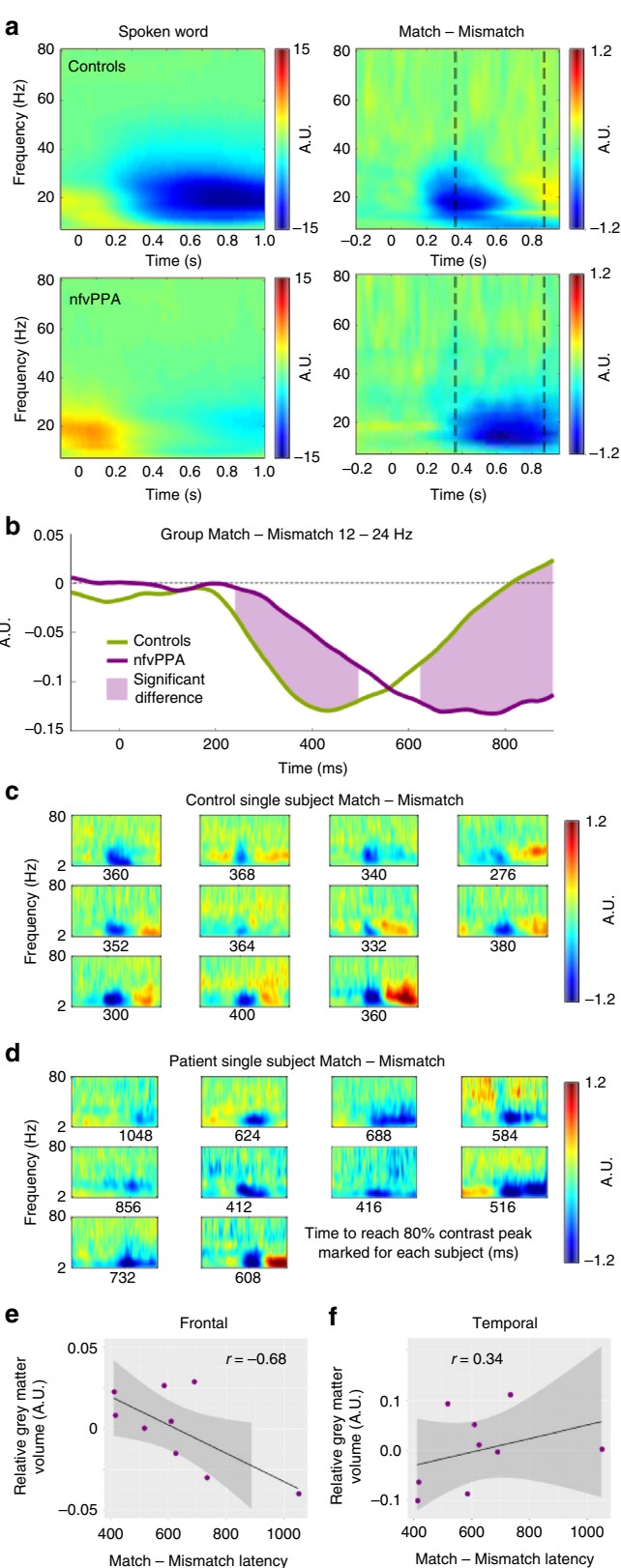

**Fig. 7** Analysis of induced responses after the spoken word. **a** Total induced power after spoken word onset and main effect of cue congruency by group. **b** Overall induced power difference between Match and Mismatch conditions in the alpha/beta overlap range. **c** Single subject time–frequency profiles for each control. The time taken to reach 80% of the peak power contrast between Match and Mismatch trials is indicated for each individual by the number below the corresponding abscissa. **d** Single subject time–frequency profiles for each patient. **e** Significant negative correlation between frontal grey matter volume (adjusted for age and total-intracranial volume at the co-ordinates in Fig. 5c) and the time taken to express a congruency contrast (**d**). The grey shaded area indicates the 95% confidence band for the regression line, marked in black. **f** No significant correlation between similarly adjusted superior temporal grey matter volume and effect latency

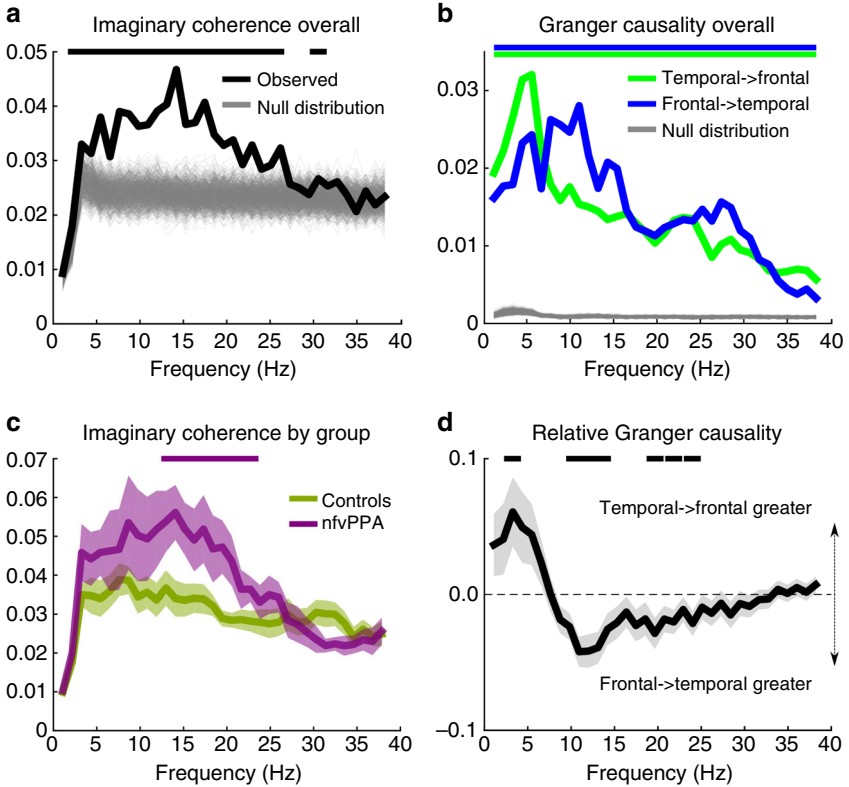

**Fig. 8** Coherence and connectivity analysis. For the time series of frontal and temporal sources of interest between 0 and 900 ms after every spoken word onset. The evoked waveform was subtracted from the time series of frontal and temporal source activity between 0 and 900 ms after every spoken word onset before analysis. Horizontal lines at the top of each plot denote frequencies at which the line of matching colour statistically exceeds either the null distribution **a** and **c**, its counterpart condition (**b**) or differs from zero (**d**). A: Imaginary coherence. The median of the observed inter-source coherence is shown in black. 1000 randomisations of the null distribution are shown in grey. **b** Imaginary coherence by group. Shading represents standard error of the mean. **c** Granger causality. Median influences from temporal to frontal sources are shown in green and frontal to temporal sources in blue. 1000 randomisations of the null distribution are shown in grey. **d** Relative normalised Granger causal relationships between temporal and frontal sources by frequency. Grey shading represents the standard error of the directionality contrast in a repeated measures general linear model

and theoretical models of predictive coding[19, 55, 56], we demonstrate that the precision of participants' predictions correlates with the magnitude of induced beta-frequency oscillations, which have recently been shown to correspond temporally and quantitatively to the updating of predictions[19].

Previous neuro-imaging evidence has suggested that frontally-mediated top-down predictions during speech perception are able to explain MEG response magnitudes[33, 40] and fMRI pattern information in the superior temporal gyrus[32]. Here, we go beyond these associations and provide causal evidence for a functional contribution of frontal networks in supporting top-down predictions by demonstrating that disruption of these networks has the remote effect of delaying the reconciliation of predictions in temporal lobe regions that are anatomically intact, with striking behavioural consequences.

Patients with nfvPPA lacked normal modulation of frontal neural activity by cue congruency, were delayed in engaging frontal regions, and showed greater frontal activity and fronto-temporal coherence than control participants. This is analogous to the observation that elderly controls globally upregulate cognitive control networks that are selectively engaged by younger listeners only when speech is degraded, and thus appear to lack difficulty-related modulation of activity[57].

In contrast to these changes, patients displayed normal power of evoked neural responses for analyses combined over sensors and had normal neural responses to changes in sensory detail. These observations provide reassurance that the lack of atrophy in auditory regions of temporal lobe does not mask a microscopic

abnormality in auditory neural function. Similarly, it is reassuring that in both groups clarity ratings were similar for mismatching and neutral cues, and the response to questionnaires suggests similar ecological perception of speech in most conditions. This excludes trivial explanations for our behavioural results such as patients becoming confused by mismatches or having an altered approach to subjective rating scales. The group difference in uncued identification of vocoded speech was small, and was accounted for in our Bayesian modelling, by using the results of Experiment 2 to individually define the precision of the sensory input for each subject and number of noise vocoder channels. Thus, our observation of abnormal effects of cue congruency in nfvPPA cannot easily be explained by basic auditory processing deficits, or by higher signal detection thresholds.

Most strikingly, patients displayed a significant delay in the effects of cue congruency in temporal lobe in our sensor space, source space and induced analyses. Under the predictive coding framework, these effects arise because the integration of prediction error is an iterative process, whereby predictions are recursively updated in the light of sensory input to minimise error[32]. For the patients with nfvPPA, the degraded neural architecture and aberrantly precise predictions might mean that this updating requires more iterations and/or that each iteration takes more time. As a consequence more frontal activity and fronto-temporal coherence is observed, and reconciliation of predictions with sensory signals is delayed.

The behavioural data, Bayesian modelling and neurophysiological results all support the proposal that perceptual predictions

operate within an hierarchical generative network, which for speech perceptions spans auditory, superior temporal, and inferior frontal cortices. Predictive coding is one framework to understand such processing, and makes a number of specific claims about the nature of top-down and bottom-up signals, which we evaluate here.

Firstly, the spatial and temporal pattern of neurophysiological responses in elderly control participants replicated those previously demonstrated in young people[40], which have been successfully modelled in a predictive coding framework[33], and which cannot be effectively modelled by sharpening theories of neural representation[32]. Current instantiations of predictive coding models state that each stage of the processing hierarchy passes forwards to the next stage only prediction errors and backwards only predictions[2, 3]. An alternative hypothesis—which might explain some of the present data—is that prior expectations are set up in frontal regions but not fed back to superior temporal regions[41, 42]. When auditory input is received, a process of sensory analysis begins in temporal lobe, with the output fed forwards to frontal regions in real-time, where a matching process occurs. If frontal regions detect that the auditory information matches expectations, they indicate that further processing is unnecessary by feeding back a stop signal to temporal regions. Such a mechanism could also account for our observation of a delayed reduction in superior temporal activity when the text cue is matching. However, this stop-signal hypothesis is unable to account for an increase in top-down, frontal to temporal, connectivity in the patient group. It is also inconsistent with fMRI evidence demonstrating an interaction of prior knowledge and sensory detail in superior temporal representations of degraded speech[32]. These fMRI findings can only by simulated by a computational model in which superior temporal regions represent the discrepancy between predicted and heard speech (i.e. prediction error). In the predictive coding model, the delayed neural effects of cue congruency observed here reflect an iterative process whereby predictions are recursively updated to minimise error; this process operates more slowly in our patient group, and is reflected in greater fronto-temporal coherence.

Most models of prediction and perception, including our own Bayesian modelling, make the assumption that perceptual outcomes represent an ideal observation of peripheral sensation. This might not be the case if individuals hold aberrant beliefs about the fidelity of their sensory input based on differences in previous experience. The results of our Bayesian modelling are inconsistent with the view that our patients are not ideal observers of their sensory experience. If patients with nfvPPA had learnt that their auditory input were unreliable, this could only explain the present data by proposing a dissociation between an underestimation of the precision of their sensory input when reporting perceptual clarity (Experiment 1) and an intact ability to discriminate sensory features when distinguishing alternative vocoded words (Experiment 2)[41]. This would manifest in our Bayesian modelling as an increase in perceptual threshold, as any given distribution of sensory input would be reported as less clear. In fact, we found that patients did not statistically differ from controls in terms of their perceptual thresholds. Indeed, the trend was towards lower thresholds, which might reflect an appropriate downwards extension of the bottom-end of their perceptual clarity rating scale to reflect the fact that they were slightly less good at identifying vocoded speech than controls (Experiment 2), indicating that they had access to slightly less sensory detail during the experiment. Therefore, our behavioural results cannot be accounted for by patients with nfvPPA not being ideal observers of vocoded speech.

The hypothesis that beta oscillations represent the instantiation of predictions has existed for some time (see ref. [56] for review). It

has been supported by evidence including computational simulations[55], and empirical observations of backward beta connectivity in speech processing[58]. More recently, direct recordings from human auditory cortex have directly linked beta frequency oscillations to the updating of predictions[19], based on correlations between observed brain activity and Bayes-optimal predictions generated from presented stimuli. In our cohort, we not only replicate the finding of beta oscillations as a correlate of prediction instantiation, but go further in demonstrating that, irrespective of disease status, the strength of this beta activity across participants relates to the precision of their predictions, as determined by our Bayesian behavioural modelling.

In clinic, patients with nfvPPA often complain of difficulties in hearing speech that are disproportionate to any measured deficits and resistant to hearing aids. We found a dissociation in subjective assessment between understanding speech in noise (which was rated as equally difficult for both groups), and speech in quiet (which only patients rated as difficult). Greater difficulty in quiet environments may seem counterintuitive at first, but it is consistent with the predictive coding hypothesis. Successful perception and comprehension of speech requires continuous updating of predictions based on sentential context and other cues. In a noisy environment, it is beneficial for listeners to rely heavily on these prior predictions as the patients do, because the sensory signal to noise ratio is poor. In a quiet environment, however, this is a suboptimal strategy as greater reliance can and should be placed on more precise or informative sensory inputs. If patients are unable to flexibly adapt the precision of their predictions to quiet listening environments, speech-in-quiet will remain difficult. It might be that globally strong predictions in nfvPPA are an adaptation to their inflexibility, as dysfunctioning networks are forced to choose one prediction strength for all scenarios. If, instead, predictions were globally weakened, this would be beneficial to the perception of speech-in-quiet but detrimental to speech-in-noise, perhaps having a greater overall cost to intelligibility in a dynamic environment (see ref. [14] for similar arguments in motor control).

We probed for subjective clarity ratings around 1050 ms after the onset of the spoken word (Fig. 2a). It could be argued that the abnormal precision of prior expectations in patients might be adaptive to the experimental context. Delayed processing might mean that they have less time for predictions to be enacted before a decision must be made, and therefore stronger predictions are required if they are to have meaningful effects. If so, our finding of increased prior precision in patients might be attenuated if clarity ratings were requested after a longer delay. However, predictions on this slower timescale would be of limited real-world consequence for speech perception and comprehension, because content-containing words such as nouns and verbs are separated by similarly short intervals in typical sentences[59], the temporal range in which human perception is optimal[60]. Visual cues from lip reading also operate over millisecond timescales and are mediated by similar increases in fronto-temporal functional connectivity to those we demonstrate here[61].

As well as explaining subjective difficulties with speech perception in nfvPPA, domain-general inflexibility in predictions could account for two other poorly understood behavioural abnormalities in this group. As shown in Supplementary Fig. 5 and Supplementary Results, we replicate the previous observation that deficits in basic auditory processing are overrepresented in patients with nfvPPA[62, 63]; in Supplementary Discussion we provide a predictive coding explanation for this in terms of inflexible priors. Secondly, agrammatism is a prominent symptom in both nfvPPA and its vascular analogue Broca-type aphasia. Patients are observed to have particular difficulties understanding complex grammatical structures containing

hierarchical structures or the passive voice[64, 65]. These structures are infrequent in daily language, and it would be reasonable to hold a prior expectation for more frequent, subject-oriented linear word orders. If patients are less able to flexibly modify this prior on the basis of violations (in this case grammatical cues), it could account for their selective behavioural impairment. This view is consistent with emerging perspectives of linguistic processing as a specialised function of a more general cognitive computational system for complex and flexible thought, based on dynamic functional interactions between inferior frontal and superior temporal cortex[66]. It is empirically supported in our group by recent evidence demonstrating that patients with frontal-lobe aphasias are impaired at detecting violations of ordering relationships in structured auditory sequences, regardless of whether those sequences are constructed of linguistic or non-linguistic stimuli[67].

## Methods

**Ethics**. All study procedures were approved by the UK National Research Ethics Service. Protocols for MEG and MRI were reviewed by the Suffolk Research Ethics Committee, and for neuropsychological tests outside of the scanner environments by the County Durham & Tees Valley Ethics Committee. All participants had mental capacity and gave informed consent to participation in the study.

**Participants**. Eleven patients with early nfvPPA were identified according to consensus diagnostic criteria[35]. The diagnosis of degenerative language disorders is complex, but nfvPPA is characterised by an aphasia with prominent apraxia of speech and/or agrammatism but without problems in single-word comprehension or object knowledge and naming. Particular care was taken to exclude patients with yes/no confusion that would confound behavioural analysis, and to include only those who lacked the lexical difficulties of logopenic and mixed aphasias. This was done in order to select patients most likely to have underlying Tau or TDP-43 related pathology preferentially involving frontal lobes (rather than Alzheimer-type pathology of parietal lobes)[39, 68–71]. On the short form of the Boston Diagnostic Aphasia Examination all patients scored 10/10 for responsive naming, 12/12 for special categories, at least 15/16 for basic word discrimination and at least 9/10 for following complex commands. One patient was unable to tolerate the MEG scanner environment so, in this specific case, contributed results only to the behavioural analysis.

We obtained standardised volumetric T1 MRI scans on nine of the patients within two months of their MEG session, which were used for co-registration and voxel-based morphometry.

Eleven age-, gender- and education-matched controls were recruited for behavioural and physiological studies (demographic information in Table 1). Thirty-six healthy age-matched control MRI datasets were selected for voxel-based morphometry.

**Voxel Based Morphometry**. Nine patients with nfvPPA underwent structural MR imaging at the Wolfson Brain Imaging Centre, University of Cambridge, UK using a 3T Siemens Magnetom Tim Trio scanner with a Siemens 32-channel phased-array head coil (Siemens Healthcare, Erlangen, Germany). A T1-weighted magnetisation-prepared rapid gradient-echo (MPRAGE) image was acquired with repetition time (TR) = 2300 ms, echo time (TE) = 2.86 ms, matrix = 192 × 192, in-plane resolution of 1.25 × 1.25 mm, 144 slices of 1.25 mm thickness, inversion time = 900 ms and flip angle = 9°. These images were compared to 36 healthy control scans with identical parameters.

All analysis was performed using SPM12 (http://www.fil.ion.ucl.ac.uk/spm). Images were first approximately aligned by coregistration to an average image in MNI space, before segmentation and calculation of total intracranial volume (TIV). After segmentation, a study-specific DARTEL template was created from the patient scans and the nine closest age-matched controls using default parameters. The remaining controls were then warped to this template. The templates were affine aligned to the SPM standard space using 'Normalise to MNI space' and the transformation applied to all individual modulated grey-matter segments together with an 8 mm FWHM Gaussian smoothing kernel. The resulting images were entered into a full factorial general linear model with a single factor having two levels, and age and TIV as covariates of no interest. This model was estimated in two ways. Firstly, a classical estimation based on the restricted maximum likelihood was performed to assess for group difference. Voxels were defined as atrophic if they were statistically significant at the peak FWE $p < 0.05$ level. Secondly, a Bayesian estimation was performed on the same model, and a Bayesian contrast between patients and controls specified. The resulting Bayesian map was subjected to hypothesis testing for the null in SPM12, resulting in a map of the posterior probability of the null at each voxel. For visualisation in Fig. 1e, this map was thresholded for posterior probabilities for the null above 0.7 and cluster volumes of greater than 1 cm³.

To assess for the dissociation between atrophic and preserved cortical regions, both model estimations were assessed at voxels of interest. Atrophic regions were assessed with classical SPM across the whole brain. This identified highly significant peaks centred in left (MNI [−37, 17, 7]) and right (MNI [37, 20, 6]) inferior frontal regions but no peaks in superior temporal regions (Supplementary Table 1). Superior temporal voxels of interest were therefore defined from the Neuromorphometrics atlas, at locations corresponding to left Heschl's gyrus within planum temporale (primary auditory cortex, MNI [−59, −24, 9]) and superior temporal gyrus (MNI [−67, −17, 3]). Frequentist probability of atrophy and Bayesian probability of no atrophy are reported at each of these four locations in results.

To create Fig. 1e, a rendering of the significant regions in each analysis, the DARTEL template images were further warped using the 'Population to ICBM Registration' function with the transformation parameters applied to all thresholded statistical maps.

To extract grey matter volume for correlation with the latency of MEG responses (Fig. 7e, f), a full factorial general linear model was constructed with the nine patients alone, with age and TIV as covariates of no interest. Each subject's age and TIV adjusted grey matter volume was extracted at the voxel closest to the MEG regions of interest (left frontal [−46 2 28]; left temporal [−56 −34 12]). A secondary SPM analysis with neural latency entered as a covariate into the model and small volume correction of 8 mm (to match the FWHM Gaussian smoothing kernel) at each location confirmed the results of the primary analysis—correlations below $p < 0.05$ were observed at the frontal but not the temporal location.

**Modifications to the Sohoglu MEG paradigm**. Stimuli and experimental procedures during neuroimaging were closely modelled on a task previously performed to evaluate influences of prior knowledge and sensory degradation in young, healthy listeners[40] (Figs. 1c, 2a). In this task, individuals are presented with a written word, followed 1050 (±50) ms later by a spoken word, which is acoustically degraded using a noise vocoder[46]. After a further delay of 1050 (±50) ms, participants are asked to rate the perceptual clarity of the vocoded word. This allows for a factorial manipulation of two stimulus dimensions that in previous studies have been shown to affect speech perception: (1) the degree of correspondence between the written and spoken words can be modified by presenting text that either matches or mismatches with the speech, (2) the amount of sensory detail can be manipulated by varying the number of channels in the noise vocoder. 108 trials of each condition were presented across six blocks. Each block contained 18 trials of each combination of vocoder channel number and cue congruency in one of two fixed random orders counterbalanced across groups. To avoid predictability, each subject observed 216 words twice in written form and twice in spoken form (once as part of a match pair and once as part of a mismatch pair) and 108 words only once (in either a match or mismatch pair). The following modifications were made to the Sohoglu et al.[40] paradigm in order to simplify procedures for patients and elderly controls. The number of channels used in the vocoder was doubled to 4/8/16, the range expected to cover the steepest portion of the psychometric response function in older adults[72]. The duration of the visual prime was increased from 200 ms to 500 ms. The resolution of the clarity rating scale was reduced from 1–8 to 1–4, so that a four-button box could be used to indicate responses. Finally, the neutral priming condition was removed to reduce the overall number of trials inside the scanner by a third, while minimising the reduction in the power with which we could test for an effect of prime congruency. 108 trials of each condition were presented across six presentation blocks. A fully crossed 2 × 3 factorial design was employed, with two levels of prime congruency (matching/mismatching) and three levels of sensory detail (4/8/16 vocoder channels). Each spoken word was presented no more than twice to each participant, once with a congruent prime and once with an incongruent prime.

**Behavioural data stimuli and procedure**. To ensure that we were observing an effect of prediction and that patients were not simply being confused by mismatching written words, experiment 1 was repeated outside the scanner with identical parameters but an additional, neutral, cue condition (Supplementary Fig. 1C), mirroring that of ref. [40]. Eighteen trials of each condition were presented in a single block.

A second experiment was undertaken to assess participants' ability to identify vocoded words. In this task, (Fig. 2c), no prior written text was provided. Participants simply heard a noise vocoded word and, 1050±50 ms after word onset, were presented with four alternatives, from which they selected the word that they had heard. We used this forced choice response format to ensure that performance was not confounded by speech production difficulties. As in Experiment 1, the clarity of the spoken word was varied between 4, 8 and 16 noise vocoder channels. The closeness of the three distractor items to the correct response was also varied by manipulating the number of neighbours between the spoken word and the alternatives. In the example shown in Fig. 2c, the spoken word is 'Gaze'. The alternatives presented to the participant comprised 'Daze' (an offset neighbour), 'Gaze' (the target), 'Game' (an onset neighbour), and 'Then' (not a neighbour). This set of four response alternatives occurred in trials where the spoken word was either: (1) "Gaze" (such that there were two word neighbours, "Daze" and "Game" in the response array), (2) "Daze" (only one onset neighbour, "Gaze"), (3) "Game" (one offset neighbour "Gaze"), or (4) "Then" (no neighbours in the response array). This achieved a factorial experimental design with a 3-level manipulation of

sensory detail fully crossed with a 4-level manipulation of distractor difficulty. There were 90 trials in a single block; at each of the three levels of sensory detail there were ten spoken words in each of cases (1) and (4) above, and five each in (2) and (3). This resulted in 30 sets of four response options each being presented three times and having three of its four members heard during the experiment. Word presentation orders were randomised across participants, but the sensory detail and written neighbour difficulty order was fixed.

Finally, we exactly replicated a subset of the tasks used by Grube et al.[62] to demonstrate peripheral auditory processing deficits in nfvPPA. We selected a cross section of tasks on which the patients with nfvPPA had displayed particular difficulties, covering a range of processing from simple to complex. These comprised pitch change detection (Grube task P1), 2 Hz and 40 Hz frequency modulation detection (Grube tasks M1 and M2), and dynamic ripple discrimination (Grube task M4)[62].

Auditory stimuli were presented in a quiet room through Sennheiser HD250 linear 2 headphones, driven by a Behringer UCA 202 external sound card, and visual stimuli were displayed on a laptop computer screen. Participants indicated responses either by pressing a number on a keyboard (clarity rating outside of MEG) or a button on a custom made response box (all other experiments).

**Behavioural data modelling**. Subjective ratings of clarity were modelled using an hierarchical Bayesian inference approach previously described for data of this type[33] (Supplementary Fig. 2). This model exploits the principles of predictive coding, in which perception arises from a combined representation of sensory input and prior beliefs[4, 5, 73]. It is able to explain both the perceptual benefit of matching prior information, as the precision of the 'posterior' (or subjective experience) is increased by congruency between the prior and the sensory input, as well as the previously observed dissociated modulations of superior temporal gyrus activity by cue congruency and sensory detail[33, 40].

The model is able to predict subjective clarity as a function of the precision of the posterior distribution, which is estimated as the precision of the sensory input multiplied by a weighted function of the precision of the prior. The weighting given to the prior information depends on its congruency. In the case of a mismatching prior, the precision of the posterior simply matches the precision of the sensory input, while for matching priors it increases as a function of the precision of prior expectation. Finally, the precision of the posterior is compared against a perceptual threshold (below which degraded speech is deemed completely unclear and given a rating of 1), and the height above this threshold is mapped to the rest of the rating scale (participants were instructed to use the full range of the rating scale, and undertook practice trials to familiarise themselves with the range of experimental stimuli—all participants were able to do this).

The precision of the sensory input was individually pre-defined for each subject at each level of sensory detail, based on their measured ability to correctly report words with that number of vocoder channels (Fig. 2d). The weighting of matching prior information was defined as 0.5, reflecting the experimental context in which 50% of written words were congruent with (i.e. matched) the degraded spoken words. In open set listening situations like those used in the present experiment, the weighting of prior expectation for the occurrence of any uncued word is the inverse of the number of nouns in the participant's lexicon. We therefore approximated the weighting to zero for both mismatching and neutral prior expectations[32]. This accounts for the observation that clarity ratings in this and previous experiments were almost identical following mismatching and neutral (uninformative) written words[27, 40].

It is possible that the patients might have an inappropriately high weighting for the written cue. In other words, they apply their prior expectations inflexibly, being unable to account for the fact that they will only be correct half the time. As we have a binary situation (the text was either fully matching or fully mismatching) we are unable to assess this possibility directly—in our model allowing the weighting of matching prior information to vary would be mathematically equivalent to allowing the precision of the prior to vary. Accordingly, we use the terms 'excessively precise priors' and 'inflexible priors' interchangeably.

The precision of the prior expectation and the level of the perceptual threshold were then individually optimised to provide the best-fit to each subject's clarity ratings by global minimisation of squared residuals (using the Global Optimisation Toolbox in MATLAB).

**Alternative data models**. It was observed that model fits were less good for the mismatch condition in some controls, with the model systematically under-predicting slope. The primary driver of this effect seemed to be a washing out of the effect of prior knowledge in the face of very clear speech for some individuals (i.e. there is less perceptual clarity benefit to prior knowledge if the auditory token is itself very clear). This has been previously observed in young healthy individuals[27, 40] and seems to result from participants nearing the upper end of their psychometric response functions and entering a region of non-linear response. That is, doubling from 8 to vocoder 16 channels results in more benefit in terms of perceptual clarity than doubling from 16 to 32 channels. Indeed, as can be observed from Fig. 2d many controls are reaching ceiling performance at reporting 16 channel vocoded speech in the absence of prior information. While we know that humans are able to detect sensory detail changes even in the face of unimpaired word identification (think, for example, of hi-fi reviews), it is not unreasonable to assume that these changes in

perceptual clarity might result in smaller rating changes than those that meaningfully improve word identification performance.

The model as originally formulated does not account for non-linearities of this type because the relative increase in the precision of the posterior distribution resulting from congruent prior expectations is modelled as being a constant function of sensory detail. In reality, congruent prior expectations are shifting the position on the psychometric response function (see ref. [27]). As we do not have experimental data for higher degrees of sensory detail (for example unprimed 24 or 32 channel speech) we cannot account for this directly. If, however, we allow the weighting of prior expectations to vary in the Match case for 16 channel speech this would simulate the effect of entering a flatter portion of the psychometric response function. Doing this did improve the model fits overall by reducing the under-prediction of slope in the controls. Although this resulted in small changes in the optimised values of the other model parameters it did not affect any of the statistical results or relationships reported in the paper (the priors are still more precise in the patients at $p < 0.01$, and their precision still significantly correlates with beta power during the instantiation of predictions with $r = -0.51$).

We compared the performance of the original model and the new model with the Akaike information criterion (corrected for small samples, AICc), which assesses whether additional model parameters sufficiently improve the information provided to account for reductions in parsimony. The simpler (original) model was favoured for 10 of the 11 patients, and 8 of the 11 controls. On average, the simpler model had an AICc 4.02 points lower than the more complex model (lower AICc scores are better).

Therefore, we report the results of the simpler model in the main text. Overall, however, we take reassurance from the fact that both models result in the same statistical relationships between the precision of prior expectations and beta power during the instantiation of predictions.

**MEG and EEG data acquisition and analysis**. An Elekta Neuromag Vectorview System was used to simultaneously acquire magnetic fields from 102 magnetometers and 204 paired planar gradiometers, and electrical potentials from 70 Ag–AgCl scalp electrodes in an Easycap extended 10–10% system, with additional electrodes providing a nasal reference, a forehead ground, and paired horizontal and vertical electrooculography. All data were digitally sampled at 1 kHz and high-pass filtered above 0.01 Hz. Head shape, EEG electrode locations, and the position of three anatomical fiducial points (nasion, left and right pre-auricular) were measured before scanning with a 3D digitiser (Fastrak Polhemus). The initial impedence of all EEG electrodes was optimised to below 5 kΩ, and if this could not be achieved in a particular channel, or if it appeared noisy to visual inspection, it was excluded from further analysis.

During data acquisition, the 3D position of five evenly distributed head position indicator coils was monitored relative to the MEG sensors (magnetometers and gradiometers). These data were used by Neuromag Maxfilter 2.2, to perform Signal Source Separation[74] for motion compensation, and environmental noise suppression.

Subsequent pre-processing and analysis was undertaken in SPM12 (Wellcome Trust Centre for Neuroimaging, London, UK), FieldTrip (Donders Institute for Brain, Cognition, and Behavior, Radboud University, Nijmegen, The Netherlands) and EEG lab (Swartz Center for Computational Neuroscience, University of California San Diego), implemented in MATLAB 2013a. Artefact rejection for eye movements and blinks was undertaken by separate independent component analysis decomposition for the three sensor types. For MEG data, components were automatically identified that were both significantly temporally correlated with contemporaneous electrooculography data and spatially correlated with separately acquired template data for blinks and eye movements. For EEG data, components spatially and temporally consistent with eye blinks were automatically identified with ADJUST[75]. These components were then projected out from the dataset with a translation matrix. Due to a technical difficulty during acquisition, one control subject had no signal recorded from two thirds of their EEG sensors, and one patient had seven sensors that failed quality control—these individuals were excluded from the EEG analysis, but included in MEG and behavioural analyses.

For evoked analysis, the data were then sequentially epoched from −500 to 1500 ms relative to speech onset, downsampled to 250 Hz, EEG data referenced to the average of all sensors, baseline corrected to the 100 ms before speech onset, lowpass filtered below 40 Hz, robustly averaged across epochs, refiltered below 40 Hz (to remove any high-frequency components introduced by the robust averaging procedure), planar gradiometer data were root-mean-square combined, all data were smoothed with a 10 mm spatial kernel and 25 ms temporal kernel before conversion to images in a window from −100 to 900 ms for statistical analysis.

For induced analysis, the de-artefacted continuous data were downsampled, re-referenced, baseline corrected as above, lowpass filtered below 100 Hz, notch filtered to exclude line noise between 48 and 52 Hz, then epoched, before being submitted to four separate time frequency decompositions by the Morlet wavelet method: two separate time windows of −500 to 1500 ms relative to written word and speech onsets were examined, with and without pre-subtraction of the condition-averaged waveform from every trial. These were robustly averaged and log rescaled compared to pre-visual baseline power in each frequency band. Morlet decomposition parameters were focused for sensitivity to low-mid frequencies, with seven wavelet cycles in a range from 4 to 80 Hz in steps of 2 Hz.

**Sensor-space evoked analysis**. For each sensor type separately, a flexible factorial design was specified in SPM12, and interrogated across all participants for main effects of prime congruency and clarity. For all sensor types, a scalp position of peak effect was defined where peak FWE $p < 0.01$. The sensor data at this scalp position was then compared across groups at every time point. A significant group×condition interaction was defined as at least seven consecutive timepoints of $p < 0.05$, exceeding the temporal smoothing induced by lowpass filtering at 40 Hz. This approach does not represent double dipping as the location of interest was defined by an orthogonal contrast[76–78], and in any case for the effect of congruency (where group×condition interactions were observed with this method), for both the planar gradiometers and the magnetometers the location of peak effect for patients alone was within 2 mm of the conjoint peak effect.

**Evoked data source reconstruction**. Source inversion methods by the sLORETA algorithm were identical to those employed by Sohoglu et al.[40], except that they were undertaken in SPM12 rather than SPM8. It was observed that the time widows of interest defined in healthy young controls were slightly earlier than the main data features in our cohort of more elderly controls, who displayed similar overall profiles to patients with nfvPPA (Supplementary Fig. 3). The time windows of interest were therefore slightly lengthened and delayed, to ensure that the main data variance was captured.

The aim of the source data analysis was to localise and explore the brain basis of the group by congruency interaction statistically demonstrated in the sensor space data. While localisation of clarity was undertaken across all individuals, and is shown in Fig. 5, in the absence of a group by clarity interaction in sensor space, no further analysis was performed on this condition.

From the previous studies in healthy young controls, it was anticipated that significant main effects of congruency would be observed in opposite directions in left frontal regions and left superior temporal gyrus. This was indeed the case, with a small left frontal region being significantly more active across the whole time window with Matching prior information, and a larger region centred on left superior temporal gyrus being significantly more active with Mismatching prior information. These peak locations were defined as voxels of interest, and the source power averaged for each condition at each location within every time window of interest for every individual. Independent, repeated measures ANOVAs were then performed in each time window. Those that demonstrated a statistically significant main effect of group or a group by congruency interaction are illustrated in Fig. 5c (the main effect of congruency was not examined, as this would represent double dipping at these voxels).

**Sensor-space induced analysis**. The primary analysis of induced data was undertaken in the planar gradiometers because of their superior signal to noise ratio for data of this kind[79]. Other sensor types were examined secondarily to check for consistency of effect, which was confirmed in all cases. Visual inspection of the time×frequency data at a variety of scalp positions revealed no clear difference in the pattern of effect (although its strength differed, as shown in Supplementary Fig. 4), so data were collapsed across all sensors for statistical comparison. A flexible factorial design was specified in SPM12 for time×frequency data across a time window of −100 to 1000 ms, and interrogated across all participants for all contrasts of interest (main effects of group, prime congruency and sensory detail, all pairwise and the three-way interaction). A second, confirmatory, analysis was performed with the condition-averaged waveform subtracted from every trial with identical statistical results, demonstrating that the effects were induced rather than evoked (we make no claims as to whether they are dynamic or structural[80]).

**Induced data source reconstruction**. The significant group by condition inter-actions observed in alpha and beta frequency bands were localised with the 'Data Analysis in Sensor Space' toolbox in SPM12. sLORETA was not available in this toolbox, so for closest comparability with the evoked reconstructions, the eLOR-ETA algorithm was used. Reconstructions used time frequency data at the frequency of maximum group×congruency interaction, ±6 Hz. Data were truncated at 50 principal components, to avoid any problems with beamforming after Signal Source Separation, which reduces the number of independent components in the data to around 70[74]. Lead fields were calculated over a window of interest from 350 ms to 900 ms, and sources reconstructed in three separate time windows of equal duration defined by the sensor space group×congruency interaction: 300–450 ms, where controls had a greater main effect of congruency; 450–600 ms, where there was no group×congruency interaction; and 600–750 ms, where patients had a greater main effect of congruency. A flexible factorial design was specified in SPM12, and the group by congruency interaction (already statistically demon-strated across the whole brain) thresholded for visualisation at uncorrected $p < 0.01$.

**Coherence and connectivity analyses**. The timeseries of the frontal ([−46, 2, 28]) and temporal ([−56, −34, 12]) sources of interest (Fig. 5b) were extracted between 0 and 912 ms after every spoken word using the function spm_eeg_inv_extract. The condition-averaged waveform (i.e. the evoked response) in each source was then subtracted from every trial to result in data with zero-mean and approximate stationarity within the time window of interest. The Fourier spectra were then computed in FieldTrip using multitapers with a ±4 Hz smoothing box. This decomposition was then subjected to separate FieldTrip connectivity analyses with either imaginary coherence or Granger causality. This same procedure was repe-ated 1000 times with the trial labels in each region shuffled to create a null dis-tribution. Statistical assessment of the presence of coherence or connectivity at each frequency involved the comparison of the observed data against the null dis-tribution (Fig. 8a, c). Between-group comparisons of imaginary coherence employed unpaired $t$-tests with unequal variance (the normality assumption was not violated), cluster corrected for multiple comparisons (Fig. 8b). To compare the strength of Granger causal relationships between regions, we first corrected for differences in signal to noise ratio between participants and regions by dividing the magnitude of each frequency value by the across-frequency mean for that individual-region pair. This created a profile of relative influence for each region at each frequency, corrected for overall differences in signal strength. At each fre-quency, the significance of 'directionality' (i.e. temporal to frontal vs frontal to temporal) was assessed with a repeated measures general linear model, and the output corrected for multiple comparisons (Fig. 8d).

**Data availability**. The processed data that support the findings of this study are available on request from the corresponding author T.E.C. The raw data are not publicly available due to file size, and because participant consent was not obtained for such data sharing, but anonymised data may also be requested for non-commercial academic research purposes. Code for the Bayesian behavioural modelling is available from https://github.com/thomascope/Bayesian_Model_Code. Code for MEG pre-processing and analysis is available from https://github.com/thomascope/VESPA/tree/master/SPM12version/Standalone%20preprocessing%20pipeline.

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

## Acknowledgements

The authors would like to acknowledge assistance with the optimisation of stimulus presentation and MEG data acquisition from Maarten Van Casteren, statistical advice from Prof. Richard Henson, neuropathological opinion from Dr. Kieren Allinson, and contributions to behavioural interpretations from Dr. Dennis Norris. Most importantly, we would like to thank the patients and their families for giving so generously and freely of their time. This study was supported by the National Institute for Health Research, the Association of British Neurologists and Patrick Berthoud Charitable Trust (TEC fellowship); the Wellcome Trust (JBR Senior Fellowship, 103838); the Evelyn Trust; and the Medical Research Council Cognition and Brain Sciences unit (MC-A060-5PQ30, MC-A060-5PQ80).

## Author contributions

The study was conceptualised by M.H.D. and J.B.R., and designed by T.E.C., E.S., K.P., C.D., M.G., R.P.C. and M.H.D. Data were collected by T.E.C. and J.W., and analysed by T.E.C. with assistance from E.S., W.S. and P.S.J. All authors contributed to interpreting the results and writing the paper.

## Additional information

**Competing interests:** The authors declare no competing financial interests

