## [Peer Review File · Nature Communications]

Reviewers' comments:

Reviewer #1 (Remarks to the Author):

This is a study of the early brain and behavioral changes in speech perception secondary to nonfluent primary progressive aphasia (nfvPPA) framed in the context of a deficit in predictive coding. The manuscript is well written and the methods are generally sound although lacking in a few details. There are a multitude of results that are coupled with Bayesian modeling to support a predictive coding explanation. The data are interesting. The take home message is that frontal lobe damage/degeneration slows down cross modal speech perception and that the frontal lobe is engaged more substantially, and performance is more negatively affected, when a degraded signal is matched with an incongruent prime. I suspect that a measure of the dynamics of the process, like interregional connectivity or coherence, would have revealed reduced coupling between frontal and temporal cortical areas. It is not clear why this was not done given that the necessary data were available. While I appreciate the attempt to use the modeling to uncover a more nuanced explanation to the data, it feels more like another level of description more so than a more informed understanding. Conceptually, framing the results in the context of deficient or dysfunctional predictive coding is one of a number of possible descriptions/explanations that fit the results. An alternative model was brought up in the Discussion (stop-signal hypothesis) and discounted but surely this is not the only alternative explanation for the data.

General Comments

Overall, the data obtained on the dysfunction associated with speech processing in this clinical population is interesting. However, the interpretation of the data and modeling are from a very simplistic perspective on the manner in which cross-modal speech perception/processing operates. In the Abstract and Introduction and it was suggested that the role of top-down predictions in supporting higher level perception remains controversial. This is an inaccurate statement. Most current models of perception are clearly integrative and involve the interaction between top-down (cognitive set) or prior state related information with bottom up sensory information. For example, is well known that visual word priming under active task conditions engage a distributed network of brain regions involving at least the frontal, temporal, parietal and occipital regions associated with orthography, phonology and semantics. The tasks used in the current study clearly engage cognitive mechanisms that involve the frontal lobe which are more actively engaged when sensory signals are degraded. As a result the observation that early degeneration of the frontal lobe leads to poorer speech perceptual processing is predictable and the fact that it can be modeled when processing is interpreted as prediction is not surprising.

There are parts of the manuscript that are in need of clarity or explanation.

Methods

1) Information on the experimental design for Experiment 1 was missing. I assumed 6 conditions (match/mismatch x 3 vocoder levels) with 108 trials for each condition totaling

648 trials that were distributed across the 6 blocks. How were the conditions distributed across each block? The ability to predict responses will be significantly influenced by the presentation scheme. Was there evidence of learning (a change in clarity ratings for matched and mismatched trials with the same vocoder channels) across the blocks?

2) For Experiment 2, there were no details on the total number of trials nor the distribution of the conditions.

3) Information on the details of the subset of tasks used by Grube was missing.

Results

1) Structural consequences—It was indicated that there was no evidence of atrophy in the left primary auditory cortex nor in the left superior temporal gyrus. Was the right primary auditory/superior temporal cortex examined?

2) It is stated that "...frontal neural degeneration does not lead to any consistent differences in the magnitude of the neural response evoked by single words sufficient to account for the interactions between group and experimental manipulation..." (Lines 120-122). This is an interpretation that does not belong in the results section and assumes knowledge of the relationship between the measured neural response and behavior. If there is evidence for this interpretation it should be included in the Discussion.

Discussion

1) Line 297-298—"Our source-space analysis demonstrates that frontal regions are working harder overall in 298 our patients (figure 5C), as a result of focal neurodegeneration in these areas (figure 1E)." Not sure how the source power differences presented in 5C demonstrate harder working frontal regions in the patients—please clarify.

2) Line 339-340—It was suggested that the results of clarity ratings were similar for mismatching cues. The principal finding of the study is that neurodegeneration of the frontal lobe resulted in delayed ability to identify a degraded auditory word following a primed written word. Moreover, when the visual prime and the vocoded word are the same, the speech is clearer for both groups than when the prime and test words are not the same. Clearly, previous information (the prime) facilitates clarity as long as prime and test words are the same. However, when the prime and test words are not the same, more information (vocoder channels) is needed to increase the clarity judgment but this is only observed for the control group. This difference was not adequately addressed nor interpreted.

3) The most significant behavioral difference in performance for the groups is that the clinical group is more disrupted by the prime especially by print when the auditory target is not the same lexical item. This deficit was maximal for clarity judgments (visual prime) and less so when the prime was auditory and the task was word identification. For the more severe deficit--How should this specific effect— an inability to override the priming effect of print when perceiving speech as an auditory object —be interpreted in the context of normal speech perception in which cross modal priming accompanies audiovisual speech perception

where the visual percept is visual motion of the articulators? That is, are the results merely a reflection of the experimental task or do they actually reflect an important characteristic of speech perception in general?

Reviewer #2 (Remarks to the Author):

This is a thought-provoking, very well written manuscript. The effect of selective neurodegeneration in the frontal lobe (but not temporal lobe auditory areas) is quite large, both behaviorally and physiologically. This is a very rare patient group, and it is exciting to see these data. A particular strength of the manuscript is the use of multiple imaging modalities with converging evidence. Below I raise a few points that I think the authors should address:

1) Even though the data are clear, the hypothesis is a bit unclear. For instance, l 61 refers to "lower-level neural mechanisms" (cochlear nucleus? brainstem?) and this is confusing. More to the point, it is unclear where the prediction for a *delayed* neural effect originates (l 91). Why could the response not have been attenuated and/or scrambled by a frequency-dependent phase response and/or delayed?

2) Stimulus details should be included, specifically, the signal processing strategy for the vocoder, the specific speech material (number of keywords? number and sexes of talkers?), the carrier signal(s), the sound presentation level (dB SPL), the sampling frequency, approximate visual angle and distance of the display. Considering the auditory nature of the experiment, a comment on audiometric function of the participants is pertinent (Table I).

3) I am somewhat concerned about the Bayesian Modelling. The model fits look mostly excellent, except for the mismatch control condition, where the model systematically under-predicts slope (Figure 2E, left panel, red curves). In the controls only, this makes the fitted clarity rating curves appear more similar for match versus mismatch than they truly are. This study hinges on the comparison of match versus mismatch across the two groups, so a revised statistical model without a systematic error in one of these four conditions is preferable.

Minor comments:

l 93 "altered or inflexible" – Vague. Aren't these two predictions quite different from each other?

l 284 across "the" left

Figure 2 F and elsewhere: "A.U." undefined.

Reviewer #3 (Remarks to the Author):

This manuscript presents research aimed at examining the effect of top-down predictive

information on the perception of degraded speech. In particular, the work aims to tackle the question of whether this top-down information is causal in driving the improvements in speech perception that come with prior information. To test this the authors have obtained an interesting data set where they have recorded MEG and EEG from both healthy controls and patients with early non-fluent primary progressive aphasia (nfvPPA). Subjects heard noise vocoded speech with different levels of intelligibility (based on number of vocoded channels) and were presented in advance with text that either did or did not match the vocoded speech. When the text matches, there is a marked improvement in perception of the vocoded speech. The key findings of the present paper are: 1) that the differences in the evoked responses to the degraded speech between match and mismatch trials peak later for the patients than for the controls; 2) that a model of subjects' behavioural responses suggests that patients apply prior expectations text with excessive precision (i.e., they overweight the text primes); 3) and that precision of the prior correlates with beta power during the interval between the presentation of the text and the hearing of the degraded word.

This paper tackles an important issue and uses a very interesting data set to do so. The work is principled and carefully conducted. However, I must admit that overall I found the manuscript quite difficult to parse and was left feeling quite conflicted about the results by the end. I think this is undoubtedly due to my simply not understanding certain aspects of the work. And I am happy to accept some "blame" for this. But I also thought that the writing of certain sections of the paper was such as to make things more complicated – or at least unclear – than perhaps they needed to be.

1) I suppose my biggest concern – or at least what I was most confused about – centers on the behavioural results. One of the key hypotheses at the core of the work is that "degeneration of top-down prediction mechanisms ... should impair perceptual function when prior knowledge and sensory input must be combined." But I don't see any clear evidence of impaired function in the present work. I mean I can see that patients have lower ratings for the mismatched vocoded speech. And I could imagine that this might relate to their overweighting (greater precision) of the prior information. But the patients are also worse at perceiving speech in quiet and in the word identification task in experiment 2. And, in light of this, maybe the larger benefit of the top down information in the matching trials could be taken as evidence that patients do better with prior information? However, this finding is not investigated directly, because, apparently "Comparing clarity ratings across groups is not...a direct measure of comparative listening difficulty." I don't quite know what the authors mean by "listening difficulty". Does this mean something other than "clarity"? If so, what does it mean? My failing to understand this logic left me uncomfortable with what followed. Namely, the authors use this notion of "listening difficulty" to justify sending their clarity ratings data through a Bayesian model along with some data from their second experiment. And, ultimately, they end up using the outputs of this model as their behavioural measures for between-group comparisons. The other experiment involved a vocoded word identification task. In this task, overall, it seems controls do better than patients. So, I guess the idea is to use experiment 2 to derive a baseline on how subjects do at processing vocoded speech at different levels. And then use this baseline to fit some model parameters to the data from experiment 1 that inform us as

to how subjects incorporate prior information. But the raw data in experiment 1 just seemed to be very quickly bypassed and we were brought to the model outputs so quickly that I became uncomfortable with the notion that we were really witnessing any real behavioural impairment in the patients. Indeed, in the discussion, the authors actually say "frontal neurodegeneration does not reduce the degree to which the brain employs contextual prior knowledge to guide lower-level speech perception". So how are we to reconcile this with the key hypothesis from the introduction?

Later in the discussion the authors also talk about "striking behavioural consequences". Not sure what they mean here. As is clear from the above, I wasn't terribly struck by any behavioural effects.

A related point – in the section on experiment 2, we were also told that "Crucially, patients performed better at identifying speech with 8 channels than controls did with 4 channels, despite having shown a greater facilitatory effect of prior knowledge across these conditions." I had absolutely no idea why this was crucial. Perhaps the authors could help me out please.

I also failed to understand why the "trend towards patients having lower perceptual thresholds" reflected "an appropriate downwards extension of the subjective clarity scale rather than a higher level introspective deficit". Why was this appropriate? Patients with nvPPA should be expected to have a lower threshold for what they think of as clear speech?? I don't understand.

A final comment on this issue of the behavioural data... in the supplementary material, the authors discuss the idea of a "weighting" for the prior that is distinct from the precision of the prior. And they seem to suggest that this weighting was defined as 0.5. But also that for mismatching trials it was set to zero. I was confused by this. And also wondered if, in fact, this weighting might be different between patients and controls, and not the precision.

2) If we do take it that the precision of the prior from the model is the appropriate dependent measure, might we not then expect some kind of correlation between it and the effects we see on the ERFs? Shouldn't the precision be even more precise for those subjects whose ERFs were more delayed? I know there are correlations between the precision and the beta power in the text-vocoded speech interval. But what about the responses to the vocoded speech themselves?

3) In the introduction the authors make it clear that they hypothesised that "frontal neurodegeneration would result in a delayed neural effect of prior context in temporal lobe speech regions." This hypothesis did not seem very well justified to me. Why would one hypothesize that an overly precise prior would lead to a delay? Rather than, say, just a smaller difference in responses between match and mismatch trials?

4) The authors state that there is no group x sensory detail interaction in Figure 3. But, in the figure there appears to be a large divergence in the magnetometer waveforms for the two groups between 600 and 800 ms (or even beyond that – we are not shown). We are told that testing was done at the peak effect, and also using SPM (and time-frequency). I take it that the SPM test included these longer latencies so? And I appreciate that the SPM analysis should control for multiple comparisons, but also be sensitive to clusters of time and space. But, I feel like maybe we need to be reassured that this late difference is definitely not significant. After all, one of the main findings in figure 4 is that the match – mismatch difference wave is larger for patients than controls in this precise interval.

5) Another observation about the data. In figure 4, I noticed that between 200 and 440 ms

the controls had larger frontal power than patients. Because of that, even though between 450 and 700 the patients had larger power, I was still somewhat surprised to see the power for patients being so much larger over the entire 0 – 900 ms interval. Is there something interesting going on between 700 and 900 ms that we are not seeing in this figure? Again, this overlaps a bit with the intervals that I was puzzled about in the previous point. These issues were important as it formed the basis for “Fronto-temporal dissociations in nvPPA” section of the discussion.

6) While I very much liked the finding that the precision of the prior correlated with the beta power in the interval between the text and the vocoded speech, I was a bit less comfortable with the beta analysis on the response to the spoken word. This was because we have previously been shown that there are significant between-group differences in the ERFs to the vocoded words. So I am worried that differences in the ERFs may be masquerading as beta band differences. And I note again that the beta differences tend to be quite late 450 – 800 ms, which, again, overlaps with the intervals above. Were there correlations between this beta band effect and the ERFs? If not, then this would give some support to the notion that these beta band effects are not actually just ERF differences.

Minor comments:

1) The authors reported VBM results for right and left frontal regions. But only (null) results for left auditory regions. Why not report the right auditory regions also given that the early speech processing hierarchy is thought to be largely bilateral?

2) Forgive my ignorance, but why eLORETA for one analysis and sLORETA for the other?

3) The word topology is incorrect (it’s a branch of mathematics). It should be topography (description of features across an area).

4) In fig 1 I would suggest changing “Evidence for no atrophy” to “No evidence of atrophy”.

5) I might suggest mentioning that there were only ten patients in the neural data in the main body of the paper.

Reviewer #4 (Remarks to the Author):

This paper provides evidence for the crucial role of frontal lobe during the predictive coding for speech perception. Choosing nvPPA patients was an excellent choice for this paper, because those patients have atrophic frontal cortex while their temporal lobe structure remains intact. Results are well consistent with the authors’ hypotheses; whereas the sensory manipulation of the auditory stimuli does not affect the temporal lobe neural responses, the level and timing of the prediction revealed differences between the patient and control groups. The effect of frontal lobe deterioration on the predictive coding was complicated but made sense; although the patient group depended on the prediction more strongly, they showed a greater delay in the reconciliation of prediction. To this reviewer’s knowledge, this finding is novel. Furthermore, because the theoretical framework of predictive coding can be easily generated to other sensory modalities, the result of this paper can be of interest for the general audience. Conclusions are convincing since the experimental design, the choice of neuroimaging method, and analysis schemes are all

appropriate. I have only a minor suggestion for the abstract; what "excessive precision" means was somewhat unclear when I started reading the manuscript. "Inflexible prediction" will be a better choice of expression.

Reviewers' comments:

Reviewer #1 (Remarks to the Author):

This is a study of the early brain and behavioral changes in speech perception secondary to nonfluent primary progressive aphasia (nfvPPA) framed in the context of a deficit in predictive coding. The manuscript is well written and the methods are generally sound although lacking in a few details. There are a multitude of results that are coupled with Bayesian modeling to support a predictive coding explanation. The data are interesting. The take home message is that frontal lobe damage/degeneration slows down cross modal speech perception and that the frontal lobe is engaged more substantially, and performance is more negatively affected, when a degraded signal is matched with an incongruent prime.

COMMENT: I suspect that a measure of the dynamics of the process, like interregional connectivity or coherence, would have revealed reduced coupling between frontal and temporal cortical areas. It is not clear why this was not done given that the necessary data were available.

RESPONSE: *We agree, and are grateful to the reviewer for this suggestion. We provide two new analyses, to reveal the changes in fronto-temporal connectivity and coherence during the resolution of predictions. This strengthens the paper, and allows us to rule out potential alternative explanations. We used two methods that give complementary information: imaginary coherence and Granger causality.*

Imaginary Coherence is immune to volume conduction effects and source spread (Nolte et al. , 2004) as well as differences in power (Bowyer, 2016), allowing us to be confident that relationships we describe are true reflections of the underlying brain dynamics. While this is a very robust measure, it is unable to robustly examine directionality, and it is recognised that some true relationships between sources will be invisible with this measure. We therefore also examine Granger Causality (Granger, 1969, Gow et al. , 2008). This allows us to look at the likely directionality of fronto-temporal interactions while still being relatively robust to volume conduction.

We show that both groups have significant fronto-temporal coherence at frequencies up to around 25 Hz (new figure 8A). In contrast to the alternate hypothesis posed by the reviewer, we show that coherence in the beta band (13-23Hz) is significantly stronger in patients than controls (figure 8B). We go on to demonstrate highly significant Granger Causal relationships between temporal and frontal regions (figure 8C). The magnitude of Granger Causal relationships can be affected by differences in signal to noise ratio between regions and between individuals (Nolte et al. , 2004). We therefore divided the magnitude of each frequency value by the across-frequency mean for each individual and region to create a profile of relative influence for each region at each frequency. This demonstrated significantly stronger temporal to frontal Granger Causal relationships at low frequencies, while frontal to temporal influences were stronger at higher, beta frequencies (figure 8D). This is reassuringly concordant with recent studies of written language comprehension (Schoffelen et al. , 2017).

Overall, our findings of increased coherence in the patient group in a frequency band where frontal to temporal influences predominate is entirely consistent with our proposal that neurodegeneration of frontal regions causes them to have to work harder to reconcile excessively precise predictions. We have introduced new sections to results, discussion and methods based on these findings.

Figure 8: Coherence and connectivity analysis for the time series of frontal and temporal sources of interest between 0 and 900ms after every spoken word onset. The evoked waveform was subtracted from every trial before analysis. Horizontal lines at the top of each plot denote frequencies at which the line of matching colour statistically exceeds either the null distribution (A and C), its counterpart group (B) or differs from zero (D). A: Imaginary coherence. The median of the observed inter-source coherence is shown in black. 1000 randomisations of the null distribution are shown in grey. B: Imaginary coherence by group. Shading represents standard error of the mean. C: Granger causality. Median influences from temporal to frontal sources are shown in green and frontal to temporal sources in blue. 1000 randomisations of the null distribution are shown in grey. D: Relative normalised Granger causal relationships between temporal and frontal sources by frequency. Grey shading represents the standard error of the directionality contrast in a repeated measures general linear model.

COMMENT: While I appreciate the attempt to use the modeling to uncover a more nuanced explanation to the data, it feels more like another level of description more so than a more informed understanding. Conceptually, framing the results in the context of deficient or dysfunctional predictive coding is one of a number of possible descriptions/explanations that fit the results. An alternative model was brought up in the Discussion (stop-signal hypothesis) and discounted but surely this is not the only alternative explanation for the data.

RESPONSE: *This is an important point that we have clarified and elaborated in the revised paper. Our study demonstrates that frontal lobes have top-down causal influences on neural activity in temporal lobe; i.e. higher-level neurodegeneration impacts on lower levels of the speech processing hierarchy. This finding necessarily implicates an hierarchical sensory processing model which includes top down (generative) models. Predictive coding is only one such hierarchical framework, yet it is one that makes a number of specific predictions about the nature of top-down and bottom-up signals, which are entirely consistent with our data. Rather than being another level of description, we believe that our computational modelling provides a compelling demonstration that predictive coding provides an adequate explanation of our findings. Although the success of these simulations does not rule out alternative accounts, it does at the very least set a standard against which other explanations must be judged.*

It is important to emphasise that our primary findings hold regardless of the underlying nature of the neural code – that frontal lobes exert top-down causal influences that alter perception by allowing flexibility of expectation. We now acknowledge at the outset that our data alone do not establish the code that is implicit in the Predictive Coding hypothesis, but set our novel findings in the context of a broader literature and the refutation of principal alternate model classes.

General Comments

COMMENT: Overall, the data obtained on the dysfunction associated with speech processing in this clinical population is interesting. However, the interpretation of the data and modeling are from a very simplistic perspective on the manner in which cross-modal speech perception/processing operates. In the Abstract and Introduction and it was suggested that the role of top-down predictions in supporting higher level perception remains controversial. This is an inaccurate statement.

RESPONSE: *Despite the evidence the reviewer gives below, with which we largely agree, a number of groups still hold that prior expectations are set up in frontal regions but are not fed back to superior temporal regions. For discussions in the domain of speech perception see, for example, (Norris et al. , 2000, McClelland et al. , 2006, McQueen et al. , 2006, Norris et al. , 2016). Others argue that observed top-down effects are merely epiphenomena, without a causal role in perception (Lotto et al. , 2009). We believe that our work is clear and unambiguous, and can resolve such controversy.*

COMMENT: Most current models of perception are clearly integrative and involve the interaction between top-down (cognitive set) or prior state related information with bottom up sensory information. For example, is well known that visual word priming under active task conditions engage a distributed network of brain regions involving at least the frontal, temporal, parietal and occipital regions associated with orthography, phonology and semantics. The tasks used in the current study clearly engage cognitive mechanisms that involve the frontal lobe which are more actively engaged when sensory signals are degraded. As a result the observation that early degeneration of the frontal lobe leads to poorer speech perceptual processing is predictable and the fact that it can be modeled when processing is interpreted as prediction is not surprising.

RESPONSE: *We agree that our findings are in keeping with the hypotheses generated from integrative accounts, and provide strong evidence for a causal role of frontal lobes in the reconciliation of predictions. The value of the behavioural model lies not only in the recapitulation of*

behavioural results (which we agree is not a surprise), but in the guidance it offers in the interpretation of the physiology (see below).

Other accounts might predict that frontal degeneration would lead to impaired perception when speech is degraded. However, any non-generative account remains insufficient as an explanation of why frontal neurodegeneration alters neural responses in (anatomically intact) temporal lobe regions that are specific to situations in which prior knowledge is used to modulate perception. Thus, we would argue that our findings provide compelling support for hypotheses generated from predictive coding and other generative perceptual accounts. We provide unprecedented evidence for a causal role of frontal lobes in the reconciliation of perceptual predictions.

COMMENT: Methods 1) Information on the experimental design for Experiment 1 was missing. I assumed 6 conditions (match/mismatch x 3 vocoder levels) with 108 trials for each condition totaling 648 trials that were distributed across the 6 blocks. How were the conditions distributed across each block? The ability to predict responses will be significantly influenced by the presentation scheme.

RESPONSE: *We have now more precisely outlined the experimental parameters as follows:*

108 trials of each condition were presented across six blocks. Each block contained 18 trials of each combination of vocoder channel number and cue congruency in one of two fixed random orders counterbalanced across groups. To avoid predictability, each subject observed 216 words twice in written form and twice in spoken form (once as part of a match pair and once as part of a mismatch pair) and 108 words only once in each form (in either a match or mismatch pair).

COMMENT: Was there evidence of learning (a change in clarity ratings for matched and mismatched trials with the same vocoder channels) across the blocks?

RESPONSE: *Perceptual learning is well established in this paradigm, and has been extensively examined in young participants (Sohoglu and Davis, 2016). We had already included a short section on this in the supplementary discussion, which we have now expanded in response to the reviewer's suggestion. We did observe increases in clarity rating consistent with perceptual learning between the MEG experiment (figure 2B) and the repetition of experiment 1 outside of the scanner (supplementary figure 1C), but these changes were small and could not account for our primary results. We now more explicitly acknowledge that they could be due to perceptual learning as well as a change in the likelihood of prior congruency as follows:*

While mismatch clarity ratings in nfvPPA for 8 and 16 channel speech were very slightly higher at this repetition than during the MEG session, which we speculate may reflect a combination of implicit learning³³ and a decrease in the likelihood of prior congruency from 50% to 33%, the large group by congruency interaction remained.

COMMENT: 2) For Experiment 2, there were no details on the total number of trials nor the distribution of the conditions.

RESPONSE: *This information has been added.*

There were 90 trials in a single block; at each of the 3 levels of sensory detail there were 10 spoken words in each of cases (1) and (4) above, and 5 each in (2) and (3). This resulted in 30 sets of four response options each being presented three times and having 3 of its 4 members heard during the experiment. Word presentation orders were randomised across participants, but the sensory detail and written neighbour difficulty order was fixed.

COMMENT: 3) Information on the details of the subset of tasks used by Grube was missing.

RESPONSE: *These experimental details were exactly replicated from the Grube study. We would like to provide more detail, but editorial policy for this journal is not to recapitulate methods published elsewhere. We reference the original study and explicitly list the tasks used as follows:*

... these comprised pitch change detection (Grube task P1), 2Hz and 40Hz frequency modulation detection (Grube tasks M1 and M2), and dynamic ripple discrimination (Grube task M4).

Results

COMMENT: 1) Structural consequences—It was indicated that there was no evidence of atrophy in the left primary auditory cortex nor in the left superior temporal gyrus. Was the right primary auditory/superior temporal cortex examined?

RESPONSE: *We restrict figure 1 to the left hemisphere for reasons of aesthetics and because this is a language task, but we now confirm in the text that these corresponding regions in the right hemisphere did not display significant atrophy.*

Significant atrophy was also observed in right inferior frontal regions (FWE $p = 0.004$; peak MNI [37, 20, 6]) but not right primary auditory cortex (FWE $p=1$ at MNI [59, -24, 9]) or superior temporal gyrus (FWE $p = 1$ at MNI [67 -17 3]).

COMMENT: 2) It is stated that “...frontal neural degeneration does not lead to any consistent differences in the magnitude of the neural response evoked by single words sufficient to account for the interactions between group and experimental manipulation...” (Lines 120-122). This is an interpretation that does not belong in the results section and assumes knowledge of the relationship between the measured neural response and behavior. If there is evidence for this interpretation it should be included in the Discussion.

RESPONSE: *We have clarified in the text that we are referring only to the neural responses. We did not mean to make any claims about behaviour. We intend only to reassure the reader that there are no large differences in overall signal magnitude that could manifest as spurious group by condition interactions.*

We have modified the text as follows:

Thus, frontal neurodegeneration does not lead to any large difference in the magnitude of the neural response evoked by single spoken words that could manifest as spurious group by condition interactions in neural activity.

Discussion

COMMENT: 1) Line 297-298—“Our source-space analysis demonstrates that frontal regions are working harder overall in 298 our patients (figure 5C), as a result of focal neurodegeneration in these areas (figure 1E).” Not sure how the source power differences presented in 5C demonstrate harder working frontal regions in the patients—please clarify.

RESPONSE: We clarify this - we were referring to the far right panel of 5C (in the previous submission), which depicts responses averaged over the entire duration of the epoch following presentation of spoken words. We agree that this was not immediately clear previously, so have added an additional label 5D to guide reader to the specific bar graphs that support this statement.

COMMENT: 2) Line 339-340—It was suggested that the results of clarity ratings were similar for mismatching cues. The principal finding of the study is that neurodegeneration of the frontal lobe resulted in delayed ability to identify a degraded auditory word following a primed written word. Moreover, when the visual prime and the vocoded word are the same, the speech is clearer for both groups than when the prime and test words are not the same. Clearly, previous information (the prime) facilitates clarity as long as prime and test words are the same. However, when the prime and test words are not the same, more information (vocoder channels) is needed to increase the clarity judgment but this is only observed for the control group. This difference was not adequately addressed nor interpreted.

RESPONSE: Line 339-340 addresses the similarity of behaviour in both groups between mismatching and neutral cues (i.e. mismatching prior knowledge did not have a penalty compared to no prior knowledge at all). This is a vital piece of evidence that patients with nfvPPA were subject to an excessive influence of prior knowledge, and were not simply being confused by mismatches. We have now re-ordered this paragraph to make it clear which piece of evidence excludes which alternative explanation of the data, and been more explicit about this finding in the results section.

COMMENT: 3) The most significant behavioral difference in performance for the groups is that the clinical group is more disrupted by the prime especially by print when the auditory target is not the same lexical item.

RESPONSE: *As noted in the previous comment, we explicitly exclude the possibility that the patients are more disrupted by mismatching prior knowledge, as responses are similar between mismatching and neutral primes (supplementary figure 1C). Rather, our Bayesian model demonstrates that they apply matching prior knowledge with excessive precision.*

COMMENT: This deficit was maximal for clarity judgments (visual prime) and less so when the prime was auditory and the task was word identification.

RESPONSE: *In the word identification task (experiment 2) there was no prime. The idea of this task was to examine the fidelity of degraded word identification in the absence of prior knowledge. We presented written text as response alternatives to avoid the challenges associated with non-fluent patients producing verbal responses. In this context, the patients performed almost as well as controls and were well above chance even in the most difficult listening conditions (figure 2D). The small differences in word identification performance between individuals was accounted for in the Bayesian modelling. This provides further evidence that there is a specific ‘top-down’ problem that cannot be simply explained by ‘bottom-up’ perceptual processing difficulties.*

COMMENT: For the more severe deficit--How should this specific effect— an inability to override the priming effect of print when perceiving speech as an auditory object —be interpreted in the context of normal speech perception in which cross modal priming accompanies audiovisual speech perception where the visual percept is visual motion of the articulators? That is, are the results merely a reflection of the experimental task or do they actually reflect an important characteristic of speech perception in general?

RESPONSE: *This is a fascinating question, which we address in detail in the discussion section ‘Relevance to cognitive functioning in neurodegenerative disease’ and the accompanying supplementary discussion. It has recently been demonstrated that visual cues from lip reading are mediated by similar increases in fronto-temporal functional connectivity to those we demonstrate here (Giordano et al. , 2017), and that predictive coding mechanisms are able to account for situations where auditory and visual speech are relatively synchronous (Schwartz and Savariaux, 2014). We argue that the experimental abnormalities demonstrated in the integration of prior expectations with sensory signals can directly explain what were previously poorly understood aspects of patients’ symptomatology. Future studies will test our hypotheses by manipulating expectations derived from running-speech factors such as grammatical context and the visual motion of the articulators.*

Reviewer #2 (Remarks to the Author):

This is a thought-provoking, very well written manuscript. The effect of selective neurodegeneration in the frontal lobe (but not temporal lobe auditory areas) is quite large, both behaviorally and physiologically. This is a very rare patient group, and it is exciting to see these data. A particular strength of the manuscript is the use of multiple imaging modalities with converging evidence. Below I raise a few points that I think the authors should address:

COMMENT: 1) Even though the data are clear, the hypothesis is a bit unclear. For instance, l 61 refers to “lower-level neural mechanisms” (cochlear nucleus? brainstem?) and this is confusing.

RESPONSE: We are grateful for the general comments. In the original version, on the line in question we were referring to hierarchical models of perception in a broad sense, but agree that this made things unclear when referred to the current specific experimental context. We have clarified this:

This is a critical and novel test for hierarchical models of perception, which motivates the following hypothesis: degeneration of top-down prediction mechanisms in frontal lobe should have a substantial impact on lower-level sensory responses in temporal lobe, and should impair perceptual function when prior knowledge and sensory input must be combined.

COMMENT: More to the point, it is unclear where the prediction for a *delayed* neural effect originates (l 91). Why could the response not have been attenuated and/or scrambled by a frequency-dependent phase response and/or delayed?

RESPONSE: Yes indeed, these are possibilities and we agree that the framing of our original hypothesis was overly precise. We have modified this to read:

...frontal neurodegeneration would result in a disrupted neural effect of prior context in temporal lobe speech regions.

COMMENT: 2) Stimulus details should be included, specifically, the signal processing strategy for the vocoder, the specific speech material (number of keywords? number and sexes of talkers?), the carrier signal(s), the sound presentation level (dB SPL), the sampling frequency, approximate visual angle and distance of the display. Considering the auditory nature of the experiment, a comment on audiometric function of the participants is pertinent (Table I).

RESPONSE: We have added this information. Audiograms for all subjects are already available in supplementary figure 1B – we have added an additional link to this figure in the same place as that to table 1 and in the table 1 legend.

COMMENT: 3) I am somewhat concerned about the Bayesian Modelling. The model fits look mostly excellent, except for the mismatch control condition, where the model systematically under-predicts slope (Figure 2E, left panel, red curves). In the controls only, this makes the fitted clarity rating curves appear more similar for match versus mismatch than they truly are. This study hinges on the comparison of match versus mismatch across the two groups, so a revised statistical model without a systematic error in one of these four conditions is preferable.

RESPONSE: This is an interesting observation, and we have applied a different model accordingly. The primary driver of this effect seems to be a ‘washing out’ of the effect of prior knowledge in the face of very clear speech for some individuals (i.e. there is less perceptual clarity benefit to prior knowledge if the auditory token is itself very clear). This has been previously observed in young healthy individuals

(Sohoglu et al. , 2012, Sohoglu et al. , 2014) and seems to result from participants nearing the upper end of their psychometric response functions and entering region of non-linear response. That is, doubling from 8 to vocoder 16 channels results in more benefit in terms of perceptual clarity than doubling from 16 to 32 channels. Indeed, as can be observed from figure 2D many controls are reaching ceiling performance at reporting 16 channel vocoded speech in the absence of prior information. While we know that humans are able to detect sensory detail changes even in the face of unimpaired word identification (think, for example, of hi-fi reviews), it is not unreasonable to assume that these changes in perceptual clarity might result in smaller rating changes than those that meaningfully improve word identification performance.

The model as originally formulated does not account for non-linearities of this type because the relative increase in the precision of the posterior distribution resulting from congruent prior expectations is modelled as being a constant function of sensory detail. In reality, congruent prior expectations are shifting the position on the psychometric response function (see (Sohoglu et al. , 2014)). As we do not have experimental data for higher degrees of sensory detail (for example unprimed 24 or 32 channel speech) we cannot account for this directly. If, however, we allow the weighting of prior expectations to vary in the Match case for 16 channel speech this would simulate the effect of entering a flatter portion of the psychometric response function. Doing this does improve the model fits as shown below. Although the values of the other model parameters do change slightly, this does not affect any of the statistical results or relationships reported in the paper (the priors are still more precise in the patients at $p < 0.01$, and this still significantly correlates with beta power during the instantiation of predictions with $r = -0.51$).

We have assessed the performance of the original model and the new model against the Akaike information criterion (corrected for small samples, AICc), which assesses whether additional model parameters sufficiently improve the information provided to account for reductions in parsimony. The simpler (original) model was favoured for 10 of the 11 patients, and 8 of the 11 controls. On average, the simpler model had an AICc 4.02 points lower than the more complex model (lower AICc scores are better).

Therefore, we retain the original model in the main text but introduce the new model in the main text and discuss it in detail in the relevant section of supplementary information. Overall, however, we take reassurance from the fact that both models result in the same statistical relationships between the precision of prior expectations and beta power during the instantiation of predictions.

The following text has been added to the main paper, and we include the discussion above in a new supplementary discussion section entitled "Alternative behaviour models":

Even better fits could be obtained for the controls by accounting for non-linearities in the effect of the increasing sensory detail on perceptual clarity beyond 16 vocoder channels, but analysis of the Akaike information criterion suggested that this increase in variance explained did not outweigh the loss of parsimony compared to the simpler model (see supplementary discussion). The simpler model (supplementary figure 2) was therefore retained, but all of the group differences and associations between model outputs and neurophysiology reported below remained significant if the complex model was used.

Minor comments:

COMMENT: I 93 “altered or inflexible” – Vague. Aren't these two predictions quite different from each other?

RESPONSE: *Yes, but we were agnostic before the study regarding the way that frontal degeneration would impact speech perception. The reviewer quite rightly pointed out that our earlier hypothesis regarding neural delay was excessively precise. Similarly our a-priori hypothesis must simply be that frontal neurodegeneration will have some effect on prior expectations, which will in turn have an effect on speech perception specifically when perception is mediated by such prior expectations. We have modified the wording of the hypothesis as follows:*

...we hypothesised that frontal neurodegeneration would disrupt the application of prior knowledge, leading to aberrant speech perception in nfvPPA patients.

COMMENT: I 284 across “the” left

RESPONSE: *Definite article now included.*

COMMENT: Figure 2 F and elsewhere: “A.U.” undefined.

RESPONSE: *A.U. stands for ‘arbitrary units’ – we have added this to the relevant figure legends.*

Reviewer #3 (Remarks to the Author):

This manuscript presents research aimed at examining the effect of top-down predictive information on the perception of degraded speech. In particular, the work aims to tackle the question of whether this top-down information is causal in driving the improvements in speech perception that come with prior information. To test this the authors have obtained an interesting data set where they have recorded MEG and EEG from both healthy controls and patients with early non-fluent primary progressive aphasia (nfvPPA). Subjects heard noise vocoded speech with different levels of intelligibility (based on number of vocoded channels) and were presented in advance with text that either did or did not match the vocoded speech. When the text matches, there is a marked improvement in perception of the vocoded speech. The key findings of the present paper are: 1) that the differences in the evoked responses to the degraded speech between match and mismatch trials peak later for the patients than for the controls; 2) that a model of subjects' behavioural responses suggests that patients apply prior expectations text with excessive precision (i.e., they overweight the text primes); 3) and that precision of the prior correlates with beta power during the interval between the presentation of the text and the hearing of the degraded word.

COMMENT: This paper tackles an important issue and uses a very interesting data set to do so. The work is principled and carefully conducted. However, I must admit that overall I found the manuscript quite difficult to parse and was left feeling quite conflicted about the results by the end. I think this is undoubtedly due to my simply not understanding certain aspects of the work. And I am happy to accept some "blame" for this. But I also thought that the writing of certain sections of the paper was such as to make things more complicated – or at least unclear – than perhaps they needed to be.

RESPONSE: In revising the manuscript, we have reorganised some sections and have paid particular attention to the phrasing of the hypotheses, so as to help guide the reader through the methods, data and interpretations.

COMMENT: 1) I suppose my biggest concern – or at least what I was most confused about – centers on the behavioural results. One of the key hypotheses at the core of the work is that "degeneration of top-down prediction mechanisms ... should impair perceptual function when prior knowledge and sensory input must be combined." But I don't see any clear evidence of impaired function in the present work. I mean I can see that patients have lower ratings for the mismatched vocoded speech. And I could imagine that this might relate to their overweighting (greater precision) of the prior information. But the patients are also worse at perceiving speech in quiet and in the word identification task in experiment 2. And, in light of this, maybe the larger benefit of the top down information in the matching trials could be taken as evidence that patients do better with prior information?

*RESPONSE: To some extent we agree. The explanation we favour is that the primary abnormality is that frontal neurodegeneration results in a delayed ability to reconcile predictions with sensory input. In response to the comments below, we now provide evidence that the delay in patients' neural manifestations of the effect of congruency is significantly correlated with frontal but not temporal grey matter density. We explore this idea in the discussion section '**Relevance to cognitive function in neurodegenerative disease**'.*

In brief, the successful perception and comprehension of speech requires continuous updating of predictions based on sentential context and other cues. Delayed processing might mean that patients

have less time for predictions to be enacted before a decision must be made, and therefore stronger predictions are required if they are to have meaningful effects. If, instead, predictions were globally weakened, this would be beneficial to the perception of speech-in-quiet but detrimental to speech-in-noise, perhaps having a greater overall cost to intelligibility in a dynamic environment. Notably, patients report speech perception difficulty as being the same in silence and in noise, while controls report a much easier time in silence but give the same difficulty ratings as patients in noise (this interaction is statistically significant).

Accordingly, we propose the primary consequence of frontal degeneration is a delay in the neural mechanism for the reconciliation of predictions that results in their inflexible application. In turn, this results in the formation of aberrantly precise priors in order to retain their effect. These aberrantly precise priors then manifest as increased beta power during the instantiation of predictions, in agreement with theoretical frameworks of predictive coding. This gives a rational (but not the only possible) framework to explain the primary finding, that frontal neurodegeneration results in excessively precise/inflexible prior expectations.

We now make this argument more simply and clearly in our conclusion.

COMMENT: However, this finding is not investigated directly, because, apparently “Comparing clarity ratings across groups is not...a direct measure of comparative listening difficulty.” I don’t quite know what the authors mean by “listening difficulty”. Does this mean something other than “clarity”? If so, what does it mean? My failing to understand this logic left me uncomfortable with what followed. Namely, the authors use this notion of “listening difficulty” to justify sending their clarity ratings data through a Bayesian model along with some data from their second experiment. And, ultimately, they end up using the outputs of this model as their behavioural measures for between-group comparisons. The other experiment involved a vocoded word identification task. In this task, overall, it seems controls do better than patients. So, I guess the idea is to use experiment 2 to derive a baseline on how subjects do at processing vocoded speech at different levels. And then use this baseline to fit some model parameters to the data from experiment 1 that inform us as to how subjects incorporate prior information. But the raw data in experiment 1 just seemed to be very quickly bypassed and we were brought to the model outputs so quickly that I became uncomfortable with the notion that we were really witnessing any real behavioural impairment in the patients.

RESPONSE: *Individual clarity ratings do not immediately allow one to compare listening difficulty across individuals, because participants were explicitly instructed to rate clarity across their own range of perceptual experience within the experiment. Consider the scenario in which one group of healthy observers were presented with 2/4/8 channel vocoded speech and another group of healthy observers were presented with 4/8/16 channel speech. If both groups are asked to rate the clarity of individual words on a four point scale covering their whole perceptual experience, both groups will produce a set of ratings that use the whole scale. A rating of ‘1’ simply means ‘one of the least clear words I heard today’, while a ‘4’ means ‘one of the most clear words I heard today’. It is obvious, however, that the two groups do not have equivalent listening difficulty – the 2/4/8 channel group is at a disadvantage compared to the 4/8/16 in this regard. We must assume that individuals are able to map their psychometric response function onto the range of their own perceptual experience.*

We therefore conducted experiment 2 to measure ‘listening difficulty’ objectively in terms of the percent of correct word identification at different levels of speech degradation. This is a relatively direct measure of the difficulty of sensory processing for participants.

The aim of the Bayesian modelling was different. The modelling allows one to disentangle the precision of prior expectations from a failure of introspection (see also our response to the comment about downwards extension of the clarity scale). We have significantly reconfigured our results section to make this clearer as follows:

It is important to note that participants were explicitly instructed to rate clarity across their own range of perceptual experience within the experiment, and were given training until they were able to do this. Comparing clarity ratings across groups is not, therefore, a direct measure of comparative listening difficulty as a rating of '1' simply means 'one of the least clear words I heard in the experiment', while a '4' means 'one of the clearest words I heard'. To fully assess the perceptual basis of our findings, with a further experiment and Bayesian modelling we assessed the elements contributing to perceptual clarity: 1) patients' and controls' ability to identify degraded spoken words and 2) participants' introspective ability to perform higher level estimation of the global precision of sensory input.

COMMENT: Indeed, in the discussion, the authors actually say “frontal neurodegeneration does not reduce the degree to which the brain employs contextual prior knowledge to guide lower-level speech perception”. So how are we to reconcile this with the key hypothesis from the introduction? Later in the discussion the authors also talk about “striking behavioural consequences”. Not sure what they mean here. As is clear from the above, I wasn't terribly struck by any behavioural effects.

RESPONSE: *Before our work, some readers might well have expected that neurodegeneration of frontal lobes would reduce an individual's ability to employ contextual prior knowledge to guide lower-level speech perception – i.e. that the effect of priming would be reduced.*

In fact, we show the opposite– the effect of priming is greatly enhanced in our patients, and this can be explained by excessively precise prior expectations. We suggest that inflexible predictions can explain all of the receptive symptoms in the frontal aphasias and potentially the motor features as well. We have significantly expanded this section of the discussion.

COMMENT: A related point – in the section on experiment 2, we were also told that “Crucially, patients performed better at identifying speech with 8 channels than controls did with 4 channels, despite having shown a greater facilitatory effect of prior knowledge across these conditions.” I had absolutely no idea why this was crucial. Perhaps the authors could help me out please.

RESPONSE: *We are glad of the opportunity to clarify this aspect of the study. A potential criticism of our interpretation of differential priming between groups is that priming simply has a larger behavioural effect when sensory detail is very low. Indeed, we see a diminution of the effect of priming in our subjects with 16 channel speech. If we had found that the patients were much worse than the controls at identifying vocoded words, this would have been a confound. However, the patients are in fact very good at reporting vocoded speech – even with only 4 vocoder channels and very difficult distractor items they are well above chance. They do slightly less well than controls overall, but with 8 channels the patients are better than the controls are with 4. Despite this, the patients have a much larger priming effect with 8 vocoder channels than the controls do with 4, demonstrating that a lower-level sensory impairment leading to a shift in the psychometric response function cannot explain our findings. We now make this clearer in the text as follows:*

Crucially, these data show that a lower-level impairment in perceiving vocoded speech cannot be the sole explanation of our finding of an increased congruency effect in nfvPPA patients. These patients performed better at identifying speech with 8 channels than controls did with 4 channels.

Yet, patients still display a much larger congruency effect for 8-channel vocoded words than controls do for 4-channel speech. Hence, the magnitude of congruency effects in clarity rating is not simply related to objective abilities at word identification, but rather reflects a difference in the mechanisms by which prior knowledge influences lower-level perceptual processing. We investigate the nature of this effect with a Bayesian perceptual model combining word report and clarity rating data.

COMMENT: I also failed to understand why the “trend towards patients having lower perceptual thresholds” reflected “an appropriate downwards extension of the subjective clarity scale rather than a higher level introspective deficit”. Why was this appropriate? Patients with nfvPPA should be expected to have a lower threshold for what they think of as clear speech?? I don’t understand.

RESPONSE: *We discuss this issue in the main paper discussion and supplementary discussion. As noted above, participants were explicitly instructed to rate clarity across their own range of perceptual experience within the experiment. Patients with nfvPPA were slightly less good at identifying vocoded speech than controls (experiment 2), indicating that they had access to slightly less sensory detail during the experiment. In this context, it would be appropriate to lower the ‘bottom-end’ of their perceptual scale, so as to reflect their experience within the experiment.*

An alternative explanation for our findings is that patients are not ideal observers of their sensory experience. This is not always easy for those unfamiliar with perceptual modelling to grasp, but the reasoning is as follows. Patients with nfvPPA may learn that their auditory inputs are unreliable. This could explain the present data if one also proposes a dissociation between an underestimation of the precision of their sensory input when reporting perceptual clarity (experiment 1) and an intact ability to discriminate sensory features when distinguishing alternative vocoded words (experiment 2). This would manifest in our Bayesian modelling as an increase in perceptual threshold, as any given distribution of sensory input would be reported as less clear. This is not what we observe – in fact the patients display a decrease in their perceptual thresholds, which is an appropriate ‘ideal observer’ response to slightly poorer ability to discriminate vocoded speech. We have sought to make this clearer in results and discussion as follows:

Most models of prediction and perception, including our own Bayesian modelling, make the assumption that perceptual outcomes represent an ideal observation of peripheral sensation. This might not be the case if individuals hold aberrant beliefs about the fidelity of their sensory input based on differences in previous experience 62. The results of our Bayesian modelling are inconsistent with the view that our patients are not ideal observers of their sensory experience. If patients with nfvPPA had learnt that their auditory input were unreliable, this could only explain the present data by proposing a dissociation between an underestimation of the precision of their sensory input when reporting perceptual clarity (experiment 1) and an intact ability to discriminate sensory features when distinguishing alternative vocoded words (experiment 2) 43. This would manifest in our Bayesian modelling as an increase in perceptual threshold, as any given distribution of sensory input would be reported as less clear. In fact, we found that patients did not statistically differ from controls in terms of their perceptual thresholds. Indeed, the trend was towards lower thresholds, which might reflect an appropriate downwards extension of the bottom-end of their perceptual clarity rating scale to reflect the fact that they were slightly less good at identifying vocoded speech than controls (experiment 2), indicating that they had access to slightly less sensory detail during the experiment. Therefore, our behavioural results cannot be accounted for by patients with nfvPPA not being ideal observers of vocoded speech.

COMMENT: A final comment on this issue of the behavioural data... in the supplementary material, the authors discuss the idea of a “weighting” for the prior that is distinct from the precision of the prior. And they seem to suggest that this weighting was defined as 0.5. But also that for mismatching trials it was set to zero. I was confused by this. And also wondered if, in fact, this weighting might be different between patients and controls, and not the precision.

RESPONSE: *Bayesian models of perception rely on all outcomes having some likelihood. In our experiment the participants were explicitly told that the written cue would be correct on half of trials. This is why it is weighted in the model at 0.5. This is an ‘open-set’ experiment where words are not repeated, so the likelihood of any other spoken word is “0.5 / (the number of words known by participants)”. As lexicon size is very large and was not experimentally assessed, we approximated this likelihood to zero.*

It is true that the patients might have an inappropriately high ‘weighting’ for the written cue. In other words, they apply their prior expectations ‘inflexibly’, being unable to account for the fact that they will only be correct half the time. As we have a binary situation (the text was either fully matching or fully mismatching) we are unable to assess this possibility – in our model allowing the weighting of matching prior information to vary would be exactly mathematically equivalent to allowing the precision of the prior to vary. This is why, throughout the manuscript, we use both of the terms ‘excessively precise priors’ and ‘inflexible priors’. We propose that inflexibility and excessive prediction characterise patients’ responses. They are rather similar concepts, but future work might attempt to disentangle them by investigating neural and behavioural responses to ‘close mismatches’ or otherwise altering the ‘ideal’ weighting of prior knowledge.

We have now explicitly acknowledged in our discussion and in our description of the Bayesian modelling that we have demonstrated prior expectations to be over precise or inflexible and explicitly address our inability to dissociate these terms.

COMMENT: 2) If we do take it that the precision of the prior from the model is the appropriate dependent measure, might we not then expect some kind of correlation between it and the effects we see on the ERFs? Shouldn’t the precision be even more precise for those subjects whose ERFs were more delayed? I know there are correlations between the precision and the beta power in the text-vocoded speech interval. But what about the responses to the vocoded speech themselves?

RESPONSE: *This is a very interesting point. In the initial analysis we did not examine the single subject timecourses. To address this question we have returned to our data in order to measure response latencies for the congruency effect in single participants.*

It was not immediately clear what the best measure of delay would be for correlations of this type. For instance, the absolute peak is not reliable because the contrast is sustained and therefore noise fluctuations may cause significant shifts in the peak time. Similarly, noise can unduly influence effects seen in a single, peak sensor or in single subject source-space data. A measure that appears to be robust is the neural delay to the first time each subject’s time-frequency effect of congruency (averaged over all sensors) reached 80% of its peak value. This allows us to estimate a consistent point on the ‘up-slope’ while being less vulnerable to noise fluctuations. To make space for this new analysis we have separated the previous figure 6 into separate figures covering the post-written word and post-spoken word time windows, and relegated the beta topography and source reconstruction figures to supplementary materials.

The new figure 7 below therefore shows each single subject's induced response to the congruency manipulation, along with the time taken to reach 80% of the peak overall power for this contrast between spoken words following matching and mismatching prior knowledge. The control delays were all tightly clustered between 275 and 400ms (figure 7C). Every single patient was delayed compared to every single control, with a range of 412-1048ms (figure 7D). Some of the responses were also much more sustained in the patient group.

As the reviewer predicts, delays were weakly anti-correlated with the standard deviation of the prior ($r=-0.37$, $p=0.1$). What we find even more interesting is that in the patient group the latency correlates with grey matter density (corrected for age and total intracranial volume) in our left frontal region of interest ($r=-0.68$, $p=0.042$; figure 7E) but not in our left temporal region of interest ($r=0.34$, $p=0.36$; figure 7F).

This supports the proposal we make above, that aberrant prior precision is perhaps an appropriate compensatory response to a delay in the neural mechanism for the reconciliation of predictions that results in their inflexible application. i.e. the primary problem is the frontal neurodegeneration, which results in a neural delay in prediction reconciliation. In turn, this results in the application of aberrantly precise priors, which manifest as increased beta power during the instantiation of predictions.

Figure 7: A: Total induced power after spoken word onset and main effect of cue congruency by group. B: Overall induced power difference between Match and Mismatch conditions in the alpha/beta overlap range. C: Single subject time-frequency profiles for each control. The time taken to reach 80% of the peak power contrast between Match and Mismatch trials is marked for each individual. D: Single subject time-frequency profiles for each patient. E: Significant negative correlation between frontal grey matter density (adjusted for age and total-intracranial volume at the co-ordinates in figure 5C) and the time taken to express a congruency contrast (panel D). F: No significant correlation between similarly adjusted superior temporal grey matter density and effect latency.

COMMENT: 3) In the introduction the authors make it clear that they hypothesised that “frontal neurodegeneration would result in a delayed neural effect of prior context in temporal lobe speech regions.” This hypothesis did not seem very well justified to me. Why would one hypothesize that an overly precise prior would lead to a delay? Rather than, say, just a smaller difference in responses between match and mismatch trials?

RESPONSE: *Indeed, as discussed in the responses to reviewer 2 this was an overly precise expression of our predictions. We have changed ‘delayed’ to ‘disrupted’.*

COMMENT: 4) The authors state that there is no group x sensory detail interaction in Figure 3. But, in the figure there appears to be a large divergence in the magnetometer waveforms for the two groups between 600 and 800 ms (or even beyond that – we are not shown). We are told that testing was done at the peak effect, and also using SPM (and time-frequency). I take it that the SPM test included these longer latencies so? And I appreciate that the SPM analysis should control for multiple comparisons, but also be sensitive to clusters of time and space. But, I feel like maybe we need to be reassured that this late difference is definitely not significant. After all, one of the main findings in figure 4 is that the match – mismatch difference wave is larger for patients than controls in this precise interval.

RESPONSE: *The SPM analysis included these longer latencies and no significant group by clarity interactions were observed here or elsewhere even at lenient statistical thresholds. This is now explicitly noted in the text:*

Neither at the scalp locations of the peak main effect, nor with a secondary SPM analysis of all scalp-time locations, nor with time-frequency analysis, were reliable group by sensory detail interactions demonstrated at any timepoint.

COMMENT: 5) Another observation about the data. In figure 4, I noticed that between 200 and 440 ms the controls had larger frontal power than patients. Because of that, even though between 450 and 700 ms the patients had larger power, I was still somewhat surprised to see the power for patients being so much larger over the entire 0 – 900 ms interval. Is there something interesting going on between 700 and 900 ms that we are not seeing in this figure? Again, this overlaps a bit with the intervals that I was puzzled about in the previous point. These issues were important as it formed the basis for “Fronto-temporal dissociations in nvPPA” section of the discussion.

RESPONSE: *Indeed, the power differences observed between 450 and 700ms continue to the end of the time window examined. We have added an extra panel covering a later time window to figure 5 and made clearer in the text that this is why there is greater frontal power in patients when responses are averaged over the entire 0-900ms interval.*

COMMENT: 6) While I very much liked the finding that the precision of the prior correlated with the beta power in the interval between the text and the vocoded speech, I was a bit less comfortable with the beta analysis on the response to the spoken word. This was because we have previously been shown that there are significant between-group differences in the ERFs to the vocoded words. So I am worried that differences in the ERFs may be masquerading as beta band differences. And I note again that the beta differences tend to be quite late 450 – 800 ms, which, again, overlaps with the intervals above. Were there correlations between this beta band effect and the ERFs? If not,

then this would give some support to the notion that these beta band effects are not actually just ERF differences.

RESPONSE: *To address this concern we repeated the whole post spoken word analysis with the condition-averaged evoked waveform subtracted from every trial. There was no change in the beta band responses, and all reported statistical relationships remained unchanged. We note this in the text prefacing the induced analysis. The new SPM confirming highly significant early and late clusters of group by congruency mismatch is shown below. As before, these clusters represent interactions of opposite directionality (see figure 7B and supplementary figure 4).*

Group X Match-Mismatch

Statistics: *p-values adjusted for search volume*

set-level		cluster-level				peak-level					Hz ms
D	C	$D_{FWE-corr}$	$q_{FDR-corr}$	k_E	D_{uncorr}	$D_{FWE-corr}$	$q_{FDR-corr}$	F	(Z_{Ξ})	D_{uncorr}	
0.063	2	0.000	0.000	131	0.000	0.000	0.001	32.31	5.13	0.000	22 900 1
		0.000	0.000	104	0.000	0.001	0.003	26.52	4.68	0.000	16 340 1

Minor comments:

COMMENT: 1) The authors reported VBM results for right and left frontal regions. But only (null) results for left auditory regions. Why not report the right auditory regions also given that the early speech processing hierarchy is thought to be largely bilateral?

RESPONSE: *We have now reported the lack of significant atrophy in right sided auditory regions. We concentrated on the left, because all of our experimental manipulations resulted in MEG response changes that were very strongly left lateralised:*

Significant atrophy was also observed in right inferior frontal regions (FWE $p = 0.004$; peak MNI [37, 20, 6]) but not right primary auditory cortex (FWE $p=1$ at MNI [59, -24, 9]) or superior temporal gyrus (FWE $p = 1$ at MNI [67 -17 3]).

COMMENT: 2) Forgive my ignorance, but why eLORETA for one analysis and sLORETA for the other?

RESPONSE: *sLORETA was chosen for consistency with previous studies of this paradigm (Sohoglu et al. , 2012, Sohoglu and Davis, 2016). However, induced source localisation was undertaken in the ‘Data Analysis in Sensor Space’ (DAiSS) toolbox in SPM12 – sLORETA is unfortunately not implemented in this toolbox. eLORETA was chosen as the closest available approximation. Please note that we have now moved the induced source localisations to supplementary information to make room for the new analyses of ERF latency, coherence and connectivity.*

We now make the difference explicit in the methods section:

The significant group by condition interactions observed in alpha and beta frequency bands were localised with the ‘Data Analysis in Sensor Space’ toolbox in SPM12. At the time of analysis sLORETA was not available in this toolbox, so for closest comparability with the evoked reconstructions, the eLORETA algorithm was used.

COMMENT: 3) The word topology is incorrect (it’s a branch of mathematics). It should be topography (description of features across an area).

RESPONSE: *Apologies. This was correct in the figures but incorrect in the text – we have corrected it throughout.*

COMMENT: 4) In fig 1 I would suggest changing “Evidence for no atrophy” to “No evidence of atrophy”.

RESPONSE: *This would not be the correct interpretation of the analyses. Our Bayesian testing for the null allows us to explicitly seek evidence for the absence of atrophy. It is different from the use of frequentist statistics (like SPM) in which absence of evidence must not be inferred as evidence of absence. We now make this more explicit in the figure legend:*

Regions coloured in red displayed consistent reductions in grey matter volume (FWE $p<0.05$). Regions coloured blue had strong evidence for normal cortical volume in nvfPPA (Bayesian probability of the null >0.7 , cluster volume $>1\text{cm}^3$). Uncoloured (grey) areas had no strong evidence for or against atrophy.

COMMENT: 5) I might suggest mentioning that there were only ten patients in the neural data in the main body of the paper.

RESPONSE: *We have made this explicit in the legend of table 1 and it is clear from the single subject data in new figure 7D*

Reviewer #4 (Remarks to the Author):

This paper provides evidence for the crucial role of frontal lobe during the predictive coding for speech perception. Choosing nfvPPA patients was an excellent choice for this paper, because those patients have atrophic frontal cortex while their temporal lobe structure remains intact. Results are well consistent with the authors' hypotheses; whereas the sensory manipulation of the auditory stimuli does not affect the temporal lobe neural responses, the level and timing of the prediction revealed differences between the patient and control groups. The effect of frontal lobe deterioration on the predictive coding was complicated but made sense; although the patient group depended on the prediction more strongly, they showed a greater delay in the reconciliation of prediction. To this reviewer's knowledge, this finding is novel. Furthermore, because the theoretical framework of predictive coding can be easily generated to other sensory modalities, the result of this paper can be of interest for the general audience. Conclusions are convincing since the experimental design, the choice of neuroimaging method, and analysis schemes are all appropriate. I have only a minor suggestion for the abstract; what "excessive precision" means was somewhat unclear when I started reading the manuscript. "Inflexible prediction" will be a better choice of expression.

RESPONSE: *We thank the reviewer for their extremely kind words. We have made the suggested change.*

References:

- Blank H, Davis MH. Prediction Errors but Not Sharpened Signals Simulate Multivoxel fMRI Patterns during Speech Perception. *PLoS Biol.* 2016;14(11):e1002577.
- Bowyer SM. Coherence a measure of the brain networks: past and present. *Neuropsychiatric Electrophysiology.* 2016;2(1):1.
- Friston K. Hierarchical models in the brain. *PLoS Comput Biol.* 2008 Nov;4(11):e1000211.
- Giordano BL, Ince RA, Gross J, Schyns PG, Panzeri S, Kayser C. Contributions of local speech encoding and functional connectivity to audio-visual speech perception. *eLife.* 2017;6.
- Gow DW, Segawa JA, Ahlfors SP, Lin F-H. Lexical influences on speech perception: a Granger causality analysis of MEG and EEG source estimates. *Neuroimage.* 2008;43(3):614-23.
- Granger CW. Investigating causal relations by econometric models and cross-spectral methods. *Econometrica: Journal of the Econometric Society.* 1969:424-38.
- Lotto AJ, Hickok GS, Holt LL. Reflections on mirror neurons and speech perception. *Trends Cogn Sci.* 2009 Mar;13(3):110-4.
- McClelland JL, Mirman D, Holt LL. Are there interactive processes in speech perception? *Trends in Cognitive Sciences.* 2006;10(8):363-9.
- McQueen JM, Norris D, Cutler A. Are there really interactive processes in speech perception? *Trends in Cognitive Sciences.* 2006;10(12):533-.
- Nolte G, Bai O, Wheaton L, Mari Z, Vorbach S, Hallett M. Identifying true brain interaction from EEG data using the imaginary part of coherency. *Clin Neurophysiol.* 2004;115(10):2292-307.
- Norris D, McQueen JM, Cutler A. Merging information in speech recognition: Feedback is never necessary. *Behav Brain Sci.* 2000;23(03):299-325.
- Norris D, McQueen JM, Cutler A. Prediction, Bayesian inference and feedback in speech recognition. *Language, cognition and neuroscience.* 2016;31(1):4-18.
- Schoffelen J-M, Hultén A, Lam N, Marquand AF, Uddén J, Hagoort P. Frequency-specific directed interactions in the human brain network for language. *Proceedings of the National Academy of Sciences.* 2017 July 25, 2017;114(30):8083-8.
- Schwartz J-L, Savariaux C. No, there is no 150 ms lead of visual speech on auditory speech, but a range of audiovisual asynchronies varying from small audio lead to large audio lag. *PLoS Comput Biol.* 2014;10(7):e1003743.
- Sohoglu E, Davis MH. Perceptual learning of degraded speech by minimizing prediction error. *Proceedings of the National Academy of Sciences.* 2016;113(12):E1747-E56.
- Sohoglu E, Peelle JE, Carlyon RP, Davis MH. Predictive top-down integration of prior knowledge during speech perception. *J Neurosci.* 2012 Jun 20;32(25):8443-53.
- Sohoglu E, Peelle JE, Carlyon RP, Davis MH. Top-down influences of written text on perceived clarity of degraded speech. *J Exp Psychol Hum Percept Perform.* 2014 Feb;40(1):186-99.

REVIEWERS' COMMENTS:

Reviewer #1 (Remarks to the Author):

I have examined the point by point response letter and the revised manuscript, and I feel that the points I raised in the previous round of review have been satisfactorily addressed.

Reviewer #2 (Remarks to the Author):

The authors have fully addressed my concerns and questions. I am pleased to recommend publication.

Reviewer #3 (Remarks to the Author):

Many thanks to the authors for their efforts in responding to my previous comments. I found the comments very helpful in better appreciating their work. And I think the changes they have made to the manuscript should help future readers to appreciate it also. I have no further concerns.